# Temperature sensitivity of DNA double-strand break repair underpins heat-induced meiotic failure in mouse spermatogenesis

Kodai Hirano [1,2,3], Yuta Nonami[1,2,3], Yoshiaki Nakamura[1,4], Toshiyuki Sato[1,5,6], Takuya Sato [7], Kei-ichiro Ishiguro [8], Takehiko Ogawa[7,9] & Shosei Yoshida [1,2✉]

Mammalian spermatogenesis is a heat-vulnerable process that occurs at low temperatures, and elevated testicular temperatures cause male infertility. However, the current reliance on in vivo assays limits their potential to detail temperature dependence and destructive processes. Using ex vivo cultures of mouse testis explants at different controlled temperatures, we found that spermatogenesis failed at multiple steps, showing sharp temperature dependencies. At 38 °C (body core temperature), meiotic prophase I is damaged, showing increased DNA double-strand breaks (DSBs) and compromised DSB repair. Such damaged spermatocytes cause asynapsis between homologous chromosomes and are eliminated by apoptosis at the meiotic checkpoint. At 37 °C, some spermatocytes survive to the late pachytene stage, retaining high levels of unrepaired DSBs but do not complete meiosis with compromised crossover formation. These findings provide insight into the mechanisms and significance of heat vulnerability in mammalian spermatogenesis.

[1] Division of Germ Cell Biology, National Institute for Basic Biology, National Institutes of Natural Sciences, 5-1 Higashiyama, Myodaiji, Okazaki 444-8787, Japan. [2] Department of Basic Biology, School of Life Science, Graduate University for Advanced Studies (Sokendai), 5-1 Higashiyama, Myodaiji, Okazaki 444-8787, Japan. [3] Japan Society for the Promotion of Science (JSPS), 5-3-1 Kojimachi, Chiyoda-Ku, Tokyo 102-0083, Japan. [4] Laboratory of Animal Breeding and Genetics, Graduate School of Integrated Sciences for Life, Hiroshima University, 1-4-4 Kagamiyama, Higashi-Hiroshima, Hiroshima 739-8528, Japan. [5] Department of Pathology, Nagoya University Graduate School of Medicine, 65 Tsurumai-cho, Showa-ku, Nagoya 466-8550, Japan. [6] Division of Molecular Pathology, Center for Neurological Disease and Cancer, Nagoya University Graduate School of Medicine, 65 Tsurumai, Showa-ku, Nagoya 466-8550, Japan. [7] Laboratory of Biopharmaceutical and Regenerative Sciences, Institute of Molecular Medicine and Life Science, Yokohama City University Association of Medical Science, 3-9 Fukuura, Kanazawa-Ku, Yokohama, Kanagawa 236-0004, Japan. [8] Department of Chromosome Biology, Institute of Molecular Embryology and Genetics (IMEG), Kumamoto University, 2-2-1 Honjo, Chuo-ku, Kumamoto 860-0811, Japan. [9] Department of Urology, Graduate School of Medicine, Yokohama City University, 3-9 Fukuura, Kanazawa-ku, Yokohama, Kanagawa 236-0004, Japan. ✉email: shosei@nibb.ac.jp

Mammalian spermatogenesis is a heat-sensitive process that occurs under low-temperature conditions[1]. In many mammalian species, the testes descend into the scrotum at a later stage after developing deep in the abdominal cavity[2]. Compared with the body core, the scrotal temperature is maintained to 2–6 °C lower by means of heat radiation from the skin surface and counter-current heat exchange between arterial and venal blood flows[3]. When testes experience temperature elevation, spermatogenesis is compromised, leading to infertility[4].

From a clinical perspective, heat vulnerability during spermatogenesis is a well-recognized problem related to a broad spectrum of risk factors underlying male infertility in humans[5]. Cryptorchidism, a pathological condition in which the testes do not descend completely but remain in the abdominal cavity or inguinal canal, causes spermatogenesis failure[4]. Varicocele, in which the heat exchange between the testicular artery and vein is affected, is also a significant cause of male infertility[4]. Testicular temperatures are increased in these conditions, whose surgical correction can improve spermatogenesis and fertility[6]. In these high-temperature situations, the stage and extent of spermatogenesis defects are variable[7], and the precise temperature dependence and mechanisms underlying such defects remain largely unresolved.

Various in vivo experiments have been used to study heat sensitivity in mammalian spermatogenesis. In a classic study using pigs, naturally occurring cryptorchid testes recovered spermatogenesis when cooled by circulating water, demonstrating that high temperatures cause spermatogenesis defects[8]. Artificial cryptorchidism, that is, surgical translocation of the testis into the abdominal cavity, is a widely used assay to study the effect of high temperatures of the body core on spermatogenesis[9–13]. In some studies, mice were exposed to elevated air temperatures[14,15].

Artificial cryptorchid testes exhibit variable defects[9–13]. In severest cases, all germ cells except undifferentiated spermatogonia are lost within a couple of weeks after surgical translocation, indicating that the transition from undifferentiated-to-differentiating spermatogonia is blocked. Such severe outcomes occur reproducibly if surgery follows a rigorous method, with the testes wrapped by the associated fat pad and positioned deep in the abdomen[13]. However, less thorough surgeries result in inconsistent testicular position and variable defects wherein, in some cases, only spermatogonia and spermatocytes remain, while round spermatids may persist in others[9–13]. A similar variability occurs in human testes with varicocele[7]. Thus, heat may affect multiple steps of spermatogenesis, possibly at different temperatures. However, the limitations of artificial cryptorchidism hinder the detailed understanding of temperature sensitivity. Foremost, the testicular temperature cannot be reliably measured or controlled. Furthermore, the effect of extratesticular factors mediated by other organs, hormones[16,17], or the nervous system cannot be excluded.

Transient heat shock, which provides consistent results in short time courses, has been used to elucidate the mechanisms of spermatogenesis heat impairment. Lower-body bathing of mice in hot water (e.g., at 43 °C for 15 min) elicits stress response pathways and acute loss of spermatocytes and spermatids[18,19]. However, such experiments cannot measure or control the actual testicular temperature or exclude extratesticular factors, either. Furthermore, it is unclear if the results of transient heating at non-physiologically high temperatures can be extrapolated to chronic exposure to temperatures around the body core.

Previously, we developed an ex vivo air-liquid interface culture of mouse testicular explants[20]. This setting supports the entire process of spermatogenesis over several weeks, leading to the production of functional sperm, albeit less efficiently than that produced in vivo. Using this culture, we investigated the chronic effect of precisely controlled temperatures on mouse spermatogenesis under conditions free from extratesticular factors.

## Results

**High temperature impairs mouse spermatogenesis ex vivo.** First, we measured the scrotal and body core temperatures of adult mice under normal conditions using minimally invasive thermometers[14]. The measured average temperatures were 34.0 °C and 37.8 °C in the scrota and body core, respectively, establishing the reported low-temperature condition in the scrota[4] (Fig. 1a and Supplementary Fig. 1a).

Based on these measurements, we evaluated the temperature dependence of spermatogenesis in ex vivo cultures[20]. Testis explants prepared from postnatal day 4 (P4) mice were cultured in an air-liquid interface (Fig. 1b) at controlled temperatures with an accuracy of ±0.2 °C. At 34 °C, the culture supports spermatogenesis following essentially the same time course as in vivo development[20]. To monitor the progression of spermatogenesis in the same explant during culture, we used Acr-GFP mice as the donors[21]. Green fluorescent protein (GFP) expression was activated in these mice during meiotic prophase I under an acrosin promoter. Precisely, GFP expression starts in pachytene spermatocytes that are in stage IV seminiferous epithelium and continue through subsequent steps of meiosis and the formation of round and elongating spermatids[22] (Fig. 1c, d).

The culture schedule used in this study is summarized in Fig. 1b. Initially, P4 testes contained only thin seminiferous tubules harboring immature spermatogonia (Fig.1b, inset). During the 5 weeks of culture, the media was changed once every week, and seminiferous tubules grew and enlarged over the periphery of the explants. As described previously[20], the Acr-GFP signal appeared in these seminiferous tubules, indicating that spermatogenesis had progressed to the late pachytene spermatocytes (stage IV or later) (Fig. 1d), and the central region became necrotic due to a limited supply of oxygen and nutrients (Fig. 1c, Supplementary Fig. 1d). Based on the Acr-GFP signal in the peripheral region observed by fluorescence microscopy, we scored spermatogenesis progression qualitatively from grade 0 to 4 once every week[20]. In grade 0 explants (GFP signal undetected), spermatogenesis did not reach the late pachytene stage. Grades 1–4 reflect the extent of spermatogenesis progression beyond this stage (Supplementary Fig. 1b). After 5 weeks, the explants were fixed and processed for hematoxylin-eosin (HE) staining and immunofluorescence (IF) to examine marker protein expression, including MVH (pan-germ cells[23]), SCP3 (spermatocytes[24]), KIT (differentiating spermatogonia[25,26]), GFRα1 (a fraction of undifferentiated spermatogonia responsible for stem cell activity[27–29]), and GATA4 (Sertoli cells[30]) (Fig. 1b, d).

At 34 °C (the scrotal temperature), the explants showed Acr-GFP fluorescence at nearly full grade from 2 to 5 weeks, as reported previously[20] (Fig. 1e and Supplementary Fig. 1c-d). In most cases, all stages of spermatogenic cells (i.e., spermatogonia, spermatocytes, and round and elongating spermatids) developed in the peripherally located tubules, albeit less densely than in vivo-grown testes[20] (Fig. 1f and Supplementary Fig. 1e). These findings are in line with the IF results, demonstrating the presence of many MVH+ germ cells, SCP3+ spermatocytes, KIT+ and GFRα1+ spermatogonia, and GATA4+ Sertoli cells (Fig. 1g and Supplementary Fig. 1f). In contrast, at 38 °C (body core temperature), no Acr-GFP fluorescence was detected throughout the culture period (Fig. 1e and Supplementary Fig. 1c-d). Consistently, we observed spermatocytes, but no round or elongating spermatids, in sections prepared at the end of the culture. The number of SCP3+ spermatocytes and MVH+ germ cells was lower than in the explants cultured at 34 °C (Fig. 1f-g and Supplementary Fig. 1d-f). In contrast, GFRα1+ and KIT+ spermatogonia and GATA4+ Sertoli cells were abundantly observed at 38 °C, as well as 34 °C (Fig. 1g). These findings indicate that spermatogenesis was

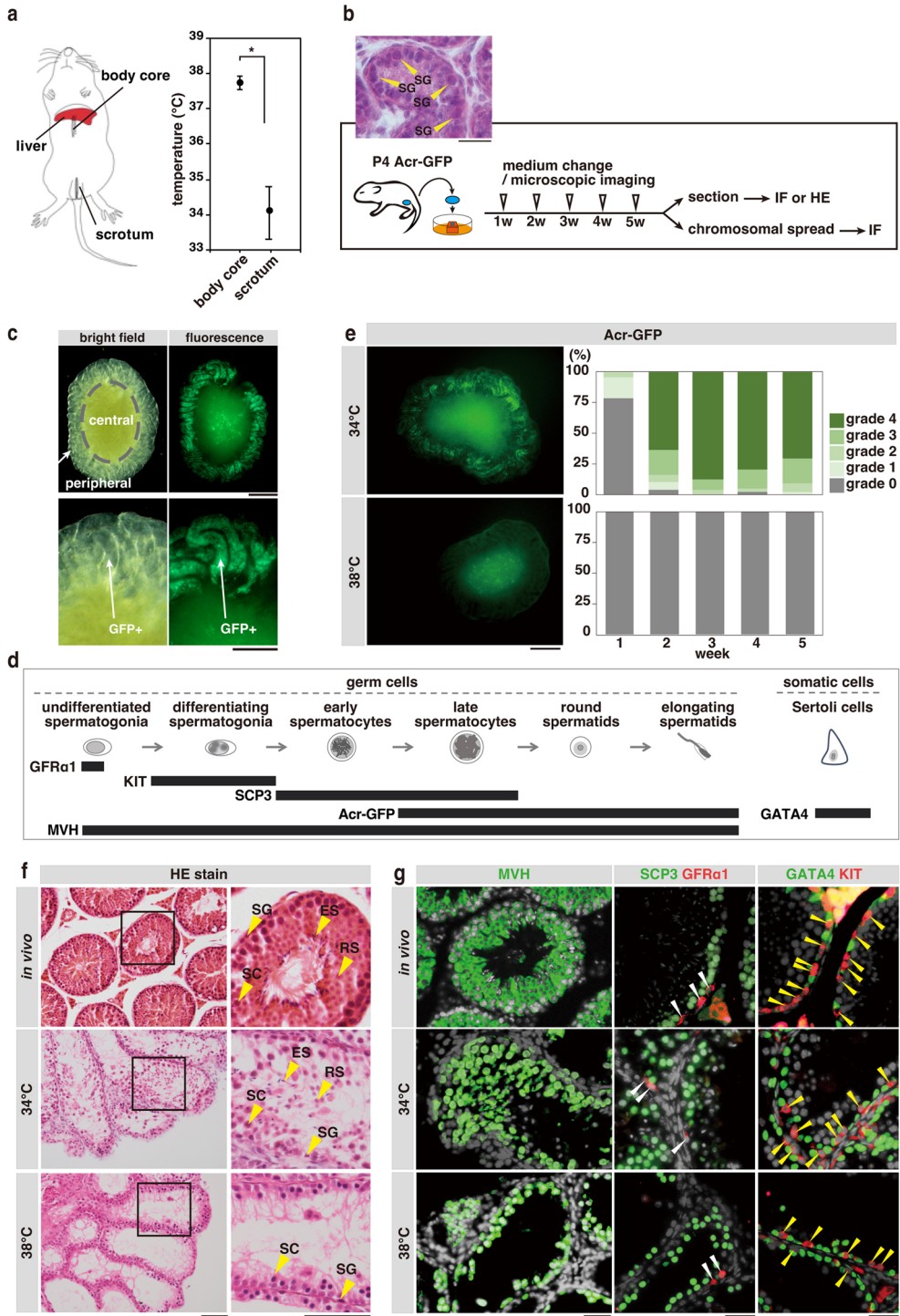

**Fig. 1 Ex vivo testis explant culture and heat sensitivity of mouse spermatogenesis. a** Scrotal and abdominal temperatures in mice measured using an implanted thermometer, shown as a schematic diagram in the left panel. Mean ± SD of body core and testis temperatures measured in 14 individuals and averaged over the times of a day are shown in the right panel (see Supplementary Fig. 1a). *$p < 0.01$. **b** Outline of the ex vivo culture experiments and a representative image of a P4 *Acr-GFP* mouse testis. Scale bar, 20 μm; SG spermatogonia, IF immunofluorescence, HE hematoxylin-eosin staining. **c** Representative fluorescence microscopic images of a testis explant harboring Acr-GFP transgene after five weeks of culture as shown in **b**. Bright filed (left) and fluorescence (right) images are shown. Scale bars; 500 and 250 μm in upper and lower panels, respectively. GFP+ seminiferous tubules were observed in the periphery while the central part was necrotic. **d** Schematic representation of mouse spermatogenesis and the expression of marker proteins used in this study. Early spermatocytes (Acr-GFP−) include leptotene, zygotene, and pachytene before stage IV, while the late spermatocytes (Acr-GFP+) include pachytene after stage IV and diplotene. **e** Acr-GFP fluorescence images of testis explants after five weeks of culture at 34 °C and 38 °C (left), and the summary of their GFP grade (0–4) over five weeks of culture at 34 °C ($N = 99$) and 38 °C ($N = 51$) (right). See also Supplementary Fig. 1c-d. Scale bar, 500 μm. **f, g** Representative images of sections of *Acr-GFP* mouse (5 week old) testes (in vivo) and the testis explants cultured for five weeks at 34 °C and 38 °C, stained with hematoxylin-eosin (**f**) and for indicated marker proteins (**g**). In (**f**), enlarged images are shown on the right. In (**g**), white and yellow arrowheads indicate GFRα1+ and KIT+ spermatogonia, respectively. SG spermatogonia, SC spermatocytes, RS round spermatids, ES elongating spermatids. Scale bars: 100 and 40 μm in (**f, g**), respectively.

impaired during the early spermatocyte stages in ex vivo culture at 38 °C.

**Temperature-dependent spermatogenesis failure at multiple steps.** To determine the temperature dependence of spermatogenesis, we incubated the explants at various temperatures ranging from 30 to 40 °C (Fig. 2a–c and Supplementary Fig. 2a). The variability in the progression of spermatogenesis between explants and within an explant prevented us from quantifying germ cells at different steps. Nevertheless, we observed reproducible and characteristic defects at specific temperatures (e.g., consistent absence of particular cell types) that are summarized in Fig. 2d and Supplementary Table 1.

We found that, among the temperatures tested, 34 °C provided the best conditions for spermatogenesis progression (Fig. 2a–d and Supplementary Fig. 2a). We observed all stages of germ cell development at 32 °C and 35 °C, although the number of elongating spermatids at these temperatures was fewer. At 36 °C, despite comparable Acr-GFP signals as at 34 °C, elongating spermatids were absent, and a few round spermatids were the most advanced germ cells (Supplementary Fig. 2b). At 37 °C, the Acr-GFP signal became weaker, no round or elongating spermatids were observed, and the Acr-GFP⁺ spermatocytes were the most advanced type of germ cells (Fig. 2a–d and Supplementary Fig. 1d-f and 2a). At 38 °C, as described earlier, no Acr-GFP signal was detected, while Acr-GFP⁻ spermatocytes and spermatogonia remained. At 39 °C, the total germ cell number was reduced but the cellular composition was similar to that at 38 °C. At 40 °C, we observed no germ cells. Massive germ cell loss occurred at 30 °C. At all temperatures, GATA4⁺ Sertoli cells densely covered the inner surface of the tubules in the periphery of the explants, emphasizing that temperature sensitivity is characteristic of germ cells.

Therefore, using ex vivo culture, we found that spermatogenesis shows a delicate temperature dependence; it does not proceed beyond the stages of early spermatocytes, late spermatocytes, and round spermatids at 38, 37, and 36 °C, respectively.

**High temperature compromises meiotic prophase I and causes apoptosis.** We then explored the process of meiotic failure at 38 °C and 37 °C. In meiotic prophase I, spermatocytes establish synapsis of homologous chromosomes and crossover, through leptotene, zygotene, pachytene, and diplotene stages[31]. To determine which stage is impaired due to high temperatures and how, we prepared chromosome spreads from explants cultured at 34, 37, and 38 °C and in vivo-grown testes to stain for SCP3 and SCP1. SCP3 is an axial element protein that is localized to unpaired chromosomes and the synaptonemal complex of paired chromosomes. SCP1 is a transverse filament protein in the central element of the synaptonemal complex, localized on only the synapsed regions of the chromosomes[32,33].

In leptotene, the axial element develops before synapsis, as detected by thin and often discontinuous SCP3 signals without overlapping SCP1. In zygotene, when homologous chromosome pairing and synapsis are established, SCP3 and SCP1 are distributed along the entire chromosome length and synapsed regions, respectively, exhibiting a characteristic Y-shape. In pachytene, when all homologues have fully synapsed and crossovers occur, SCP1 and SCP3 colocalize on the synaptonemal complex over the entire length of paired chromosomes. In diplotene, when the synaptonemal complex disassembles, partially synapsed homologue pairs with characteristic telomeric thickening of SCP3 staining were observed. Such localization of SCP3 and SCP1 was consistently observed in spermatocytes developed ex vivo at all tested temperatures (Fig. 3a). Further, co-

staining with HORMAD1, an axial element protein associated with the unsynapsed region but not with the synapsed region, verified the validity of SCP3/SCP1 staining in these samples[34] (Supplementary Fig. 3a-b).

Using these classification criteria, we examined the composition of leptotene, zygotene, pachytene, and diplotene spermatocytes in ex vivo-grown testis explants (Fig. 3a, b). As expected, the explant cultured at 34 °C contained all of these stages, with proportions similar to the in vivo control. However, at 37 °C, only a small fraction of spermatocytes reached the pachytene and diplotene stages (Fig. 3b). At 38 °C, pachytene and diplotene spermatocytes were essentially absent, although leptotene and zygotene spermatocytes were preserved (Fig. 3a, b). These findings are in line with the undetectable and weak Acr-GFP signals at 38 and 37 °C, respectively (Fig. 2a).

We then investigated if and how spermatocytes are lost when culture temperature was raised (Fig. 3c and Supplementary Fig. 3c-d). After 2 weeks of culture at 34 °C, the explants showed a prominent Acr-GFP signal, indicating that spermatogenesis proceeded to late pachytene. Consistently, these explants harbored many SCP3⁺ spermatocytes. However, over 4 days after the temperature shift to 38 °C, the Acr-GFP signal disappeared, and the number of SCP3⁺ spermatocytes reduced considerably (Fig. 3c). During this period, cleaved Caspase-3, a specific indicator of apoptosis[35], was detected in the SCP3⁺ spermatocytes (Fig. 3c and Supplementary Fig. 3d).

In summary, at 38 °C, spermatocytes do not complete chromosome pairing and die through apoptosis. At 37 °C, while many spermatocytes die, some survive until the pachytene and diplotene stages, but do not finish meiotic divisions to develop into spermatids.

**DNA double-strand breaks increase at 37 °C and 38 °C.** We further investigated the mechanisms by which meiotic prophase I failed. In the normal process (Fig. 4a), numerous DNA double-strand breaks (DSBs) are generated by the SPO11-TOPO6BL complex in the leptotene stage[31,36]. Then, RPAs are loaded to the DSBs by binding to the single-stranded DNA generated by end resection, to which the recombination proteins RAD51 and DMC1 assemble to form recombination foci[37–42]. In zygotene, RAD51 and DMC1 mediate homology-dependent DNA repair to establish synapsis between the homologous chromosomes. Chromosome pairing occurs synchronously, and its termination defines pachytene initiation. In pachytene, a minority fraction of DSBs gives rise to crossovers through a specific pathway of homologue-dependent DSB repair involving MLH1, such that each chromosome pair typically contains one (occasionally two or more) crossover(s)[43]. Crossover forms chiasmata as homologues detach in diplotene.

To determine how these processes are affected at different temperatures, we first evaluated the levels of DSBs by staining for phosphorylated histone H2AX (γH2AX)[44]. In leptotene and zygotene, newly formed DSBs elicit γH2AX generation, a process mediated by ATM[45]. γH2AX then decreases as the DSBs are repaired. Following this first wave, the second wave of γH2AX generation also occurs at later stages in response to unsynapsed chromosomes, known as the meiotic silencing of unsynapsed chromatin (MSUC)[46]. In in vivo samples, as reported, intense γH2AX signals were detected in the nuclei of leptotene and zygotene spermatocytes, but the signals were greatly reduced in pachytene, except for the intense signal in the XY-body reflecting the MSUC[44] (Fig. 4b). These observations are consistent with the results of γH2AX quantification (Fig. 4c).

In testes grown ex vivo at 34 °C, γH2AX staining was similar to that of the in vivo control (Fig. 4b). At 37 °C, γH2AX signals were

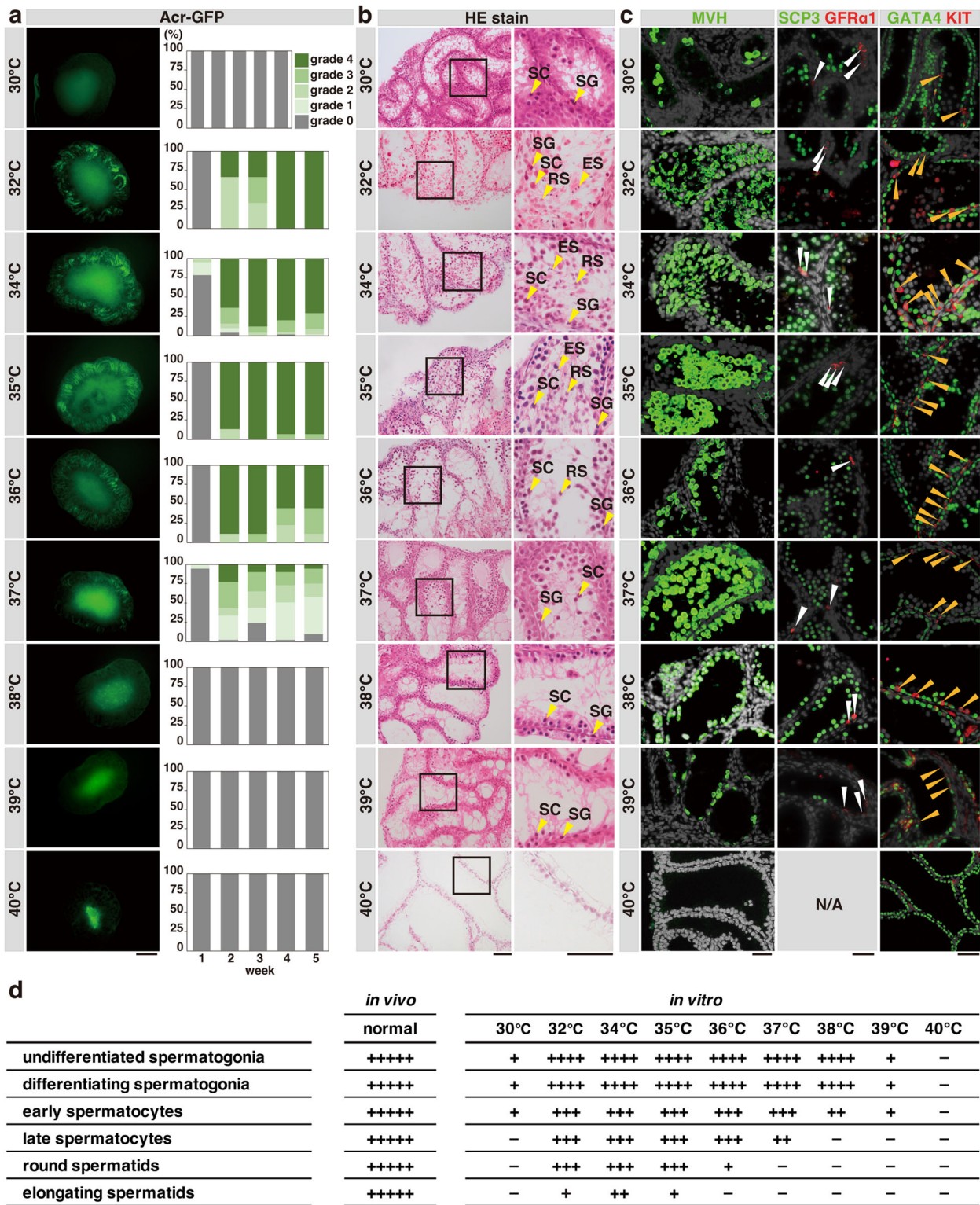

**Fig. 2 Spermatogenesis progression in testis explants cultured at a range of temperatures. a** Representative Acr-GFP fluorescence images of testis explants after 5 weeks of culture at the indicated temperatures (left) and the summarized time course of Acr-GFP grades (right). $N = 6$, 4, 15, 9, 41, 11, and 11 for 30, 32, 35, 36, 37, 39, and 40 °C, respectively. Scale bar, 500 μm. **b** Representative images of HE-stained sections of testis explants cultured at the indicated temperatures. Rectangles in the left panels indicate the area magnified on the right. Scale bars, 100 μm. SG spermatogonia, SC spermatocytes, RS round spermatids, ES elongating spermatids. **c** Representative images of testis explant sections cultured at the indicated temperatures and immunostained for the indicated proteins. White and orange arrowheads indicate GFRα1+ and KIT+ spermatogonia, respectively. Scale bars, 40 μm. Some panels in **a–c**) are also shown in Fig. 1e, f, and g, respectively. **d** Summary of the qualitative evaluation of spermatogenic cell types observed in the testes explants cultured for five weeks at the respective temperatures. −, never observed; +, observed infrequently but reproducibly; ++, found easily in some peripheral tubules; +++, observed in most peripheral tubules as a group of cells; ++++, always observed as a robust population close to in vivo samples; +++++, observed as a completely developed population at levels of in vivo-developed testis, which never occurred in our ex vivo cultures.

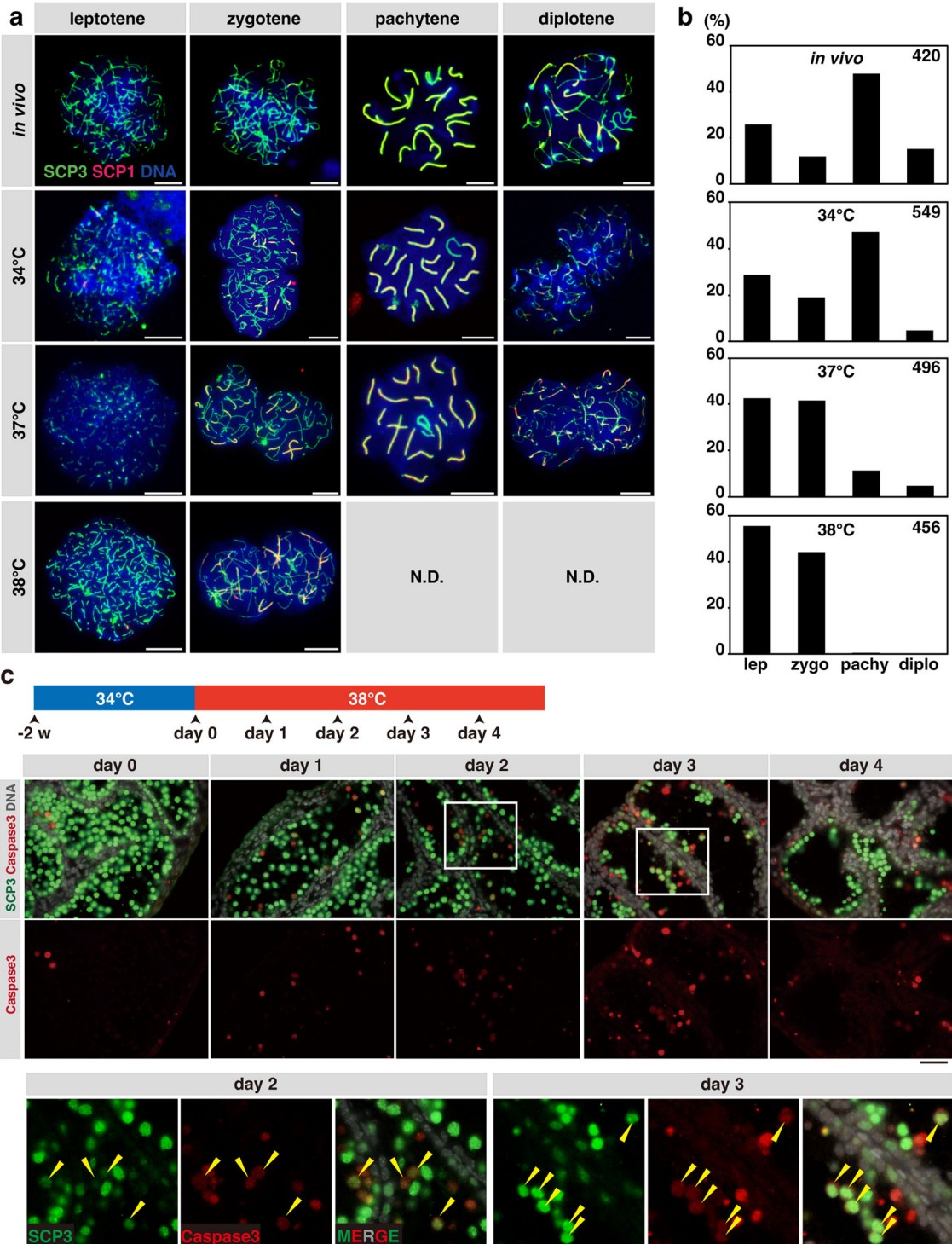

**Fig. 3 Meiotic prophase I progression and apoptosis in spermatocytes in ex vivo culture at different temperatures. a** Representative images of chromosomal spreads at different stages of meiotic prophase I, prepared from in vivo-developed testes of 5-week-old *Acr-GFP* mice and from testis explants cultured at 34, 37, or 38 °C for 5 weeks as indicated. Samples were stained for SCP3, SCP1, and DNA (Hoechst 33342). Scale bars, 10 μm. **b** Proportions of spermatocytes in the leptotene (lep), zygotene (zygo), pachytene (pachy), and diplotene (diplo) stages, found in explants cultured for 5 weeks at the indicated temperatures classified visually on the chromosome spreads after immunofluorescence-staining for SCP1 and SCP3 as (**a**), according to the criteria described in the text. Values obtained from chromosome spreads prepared from pooled testicular cells of two in vivo-developed testes from different individuals and those from 6 to 7 ex vivo-grown explants are summarized. The total number of spermatocyte nuclei counted is indicated at the upper right of each panel. Percentages of spermatocytes nuclei in which all the autosomes have completed synapsis (i.e., the sum of pachytene and diplotene spermatocytes) were 63, 52, 16, and 0.5% for in vivo and ex vivo samples at 34, 37, and 38 °C, respectively. **c** Detection of cleaved Caspase-3 (red) and SCP3 (green) in testis explants following the temperature shift from 34 to 38 °C. Double-stained images overlaid with DNA staining (gray) and the signals for cleaved Caspase-3 alone are shown in the upper and lower panels, respectively. Enlarged images at positions indicated by rectangles are also shown below. Yellow arrowheads, Caspase-3+/SCP3+ double-positive cells (the dying spermatocytes). Scale bars, 40 μm.

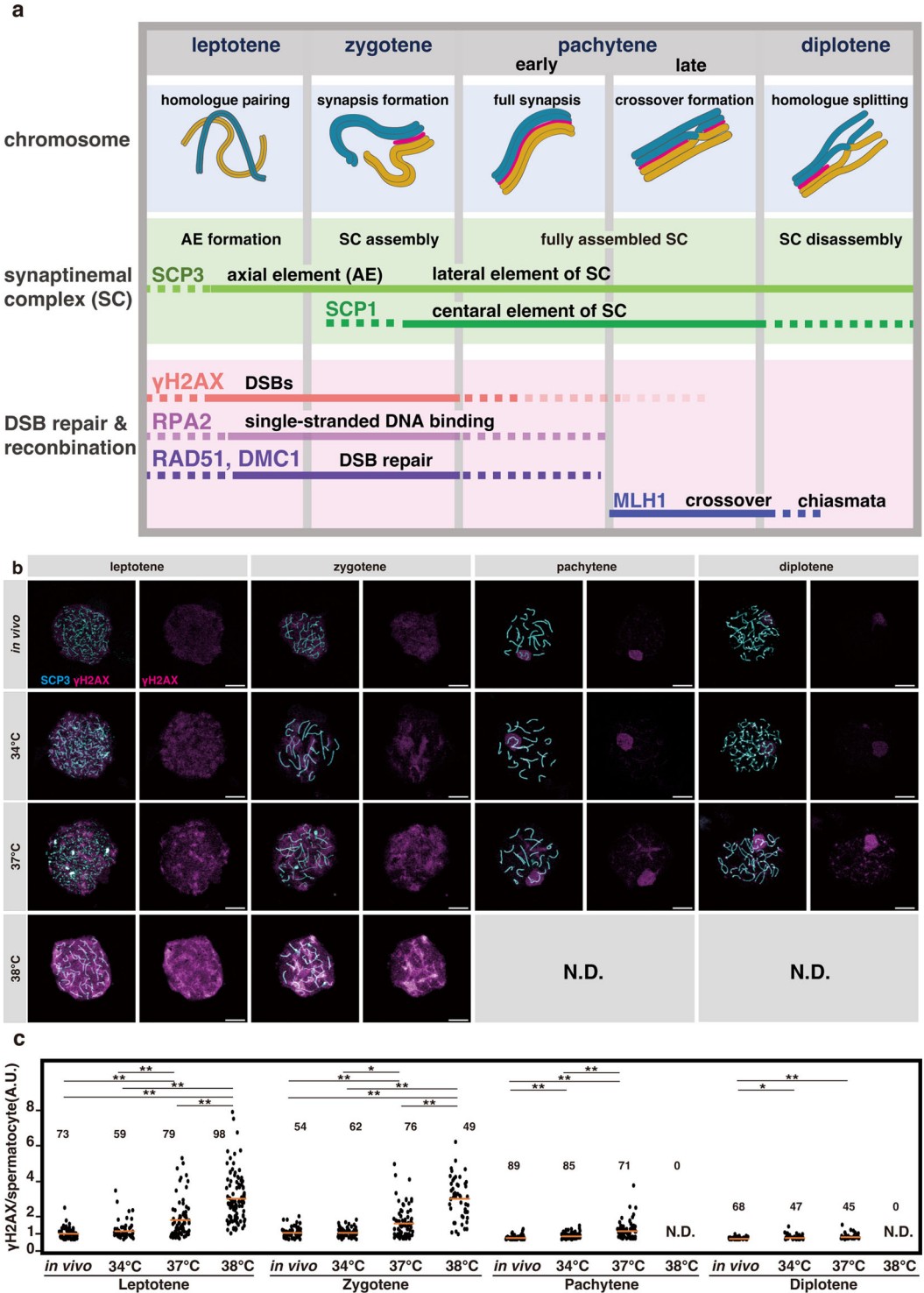

**Fig. 4 Events in meiotic prophase I and evaluation of DNA double-strand breaks (DSBs) at different temperatures. a** Schematic representation of the stages of progression and events during mouse meiotic prophase I, and the expression of key proteins assayed in this study. Early pachytene includes those before stage IV of seminiferous epithelium, while those at stage IV or later are designated as late pachytene. **b** Representative images of chromosome spreads of spermatocytes in meiotic prophase I substages prepared from the testes of 5-week-old *Acr-GFP* mice and testis explants cultured for five weeks at 34, 37, or 38 °C. Samples were stained for γH2AX (magenta) with SCP3 (cyan) used for staging. The γH2AX signals are also shown separately in the right panels. Scale bars, 10 µm. **c** Dot plots showing the levels of the γH2AX signal in each spermatocyte nucleus at the indicated substage quantified from immunofluorescence images as shown in (**b**). Each dot represents an individual spermatocyte with horizontal bars indicating the mean values. The number of spermatocytes analyzed are indicated above the plots (N.D. not detected). Cells prepared from four testes from different individuals, and 14, 15, and 15 explants cultured at 34, 37, and 38 °C, respectively, were pooled for each sample. Multiple comparisons were conducted using the Kruskal–Wallis test among all values within each category and gave *p*-values of 2.31E-34, 1.13E-19, 2.53E-19, and 3.76E-5 for leptotene, zygotene, pachytene, and diplotene, respectively, and using the Steel–Dwass test for combinations of two values within each category resulting in the *p*-values as indicated in the plots (\**p* < 0.05, \*\**p* < 0.01).

higher than at 34 °C in leptotene and zygotene. In the few surviving pachytene and diplotene spermatocytes, weak γH2AX signals remained on and outside the chromosomal axes (Fig. 4b). In agreement with this, the quantified levels of γH2AX were highly variable in leptotene and zygotene spermatocytes, ranging from being comparable to being greater than that at 34 °C and in vivo; and the pachytene and diplotene spermatocytes showed higher γH2AX levels than that at 34 °C and in vivo (Fig. 4c). In addition to the unrepaired DSBs, such abundant γH2AX levels in pachytene may also be due to that generated through the MSUC. At 38 °C, γH2AX increased further in leptotene and zygotene, while no pachytene or diplotene spermatocytes were observed (Fig. 4b, c). To summarize, DSBs levels increase at 38 °C and, to a lesser extent, at 37 °C, implying enhanced generation or compromised repair.

**High temperature compromises the DSB repair machinery**. To determine whether DSB repair was impaired at high temperatures, we examined the localization of RPA2, RAD51, and DMC1[37–42].

In in vivo samples, as reported, RPA2 foci increased in late leptotene and zygotene, followed by a decrease in pachytene as DSB repair proceeded (Fig. 5a, b)[39,41,47]. In explants grown at 34 °C, the number of RPA2 foci was slightly smaller than in vivo samples. A larger decrease was observed at 37 °C, whereas comparable numbers of RPA2 to that in in vivo samples were observed at 38 °C (Fig. 5a, b). However, given the considerable increase in DSBs at 37 °C and the even more dramatical increase at 38 °C (Fig. 4b, c), we concluded that the loading of RPA2 to the DSB sites is compromised at 37 °C and 38 °C. Specifically, these temperature conditions may compromise the RPA binding to the single-stranded DNA, or the generation of single-stranded DNA by end resection[48].

Next, we stained for RAD51 and DMC1 (Fig. 6a–d). In in vivo samples, we detected these proteins at many foci (ranging from about 50–250) in late leptotene and zygotene spermatocytes; their numbers decrease in pachytene as DSB repair proceeds, as reported previously[49–52]. In explants cultured at 34 °C, they were found in numbers similar to that of the in vivo control, although the number of RAD51 foci was moderately increased (Fig. 6a–d). However, at 38 °C, we observed remarkably fewer RAD51 and DMC1 foci in late leptotene and zygotene spermatocytes, a fraction of which harbored abnormally few foci (<50) (Fig. 6a–d). At 37 °C, RAD51 and DMC1 foci numbers were slightly reduced in late leptotene and zygotene; the few surviving pachytene spermatocytes showed similar numbers to that of the in vivo control (Fig. 6a–d). In summary, the recruitment and assembly of RAD51 and DMC1 to form recombination foci are compromised at 37 °C and 38 °C. These defects must contribute to the abundant DSBs observed at high temperatures.

**High temperature causes asynapsis between homologous chromosomes**. Abundant DSBs can interfere with the establishment of synapses between homologous chromosomes[48,53]. Therefore, we investigated chromosome pairing in spermatocytes grown at various temperatures. In normal spermatogenesis, chromosome pairing occurs synchronously. Accordingly, a spermatocyte is classified as a zygotene, if its nucleus contains both synapsed and unsynapsed regions (spanning over the entire or partial lengths of the chromosomes), a criterion which we followed in this study (Figs. 3–6).

However, in explants grown at 37 °C and 38 °C, we found that chromosome pairing was aberrantly disordered in some spermatocytes fulfilling the aforementioned criteria for zygotene. In these cells, while some chromosomes had synapsed completely, others remained unsynapsed at all or were largely unsynapsed (Fig. 7a, b

and Supplementary Fig. 4a). In addition, we occasionally observed the pairing of non-homologous chromosomes, which is another consequence of asynapsis (Fig. 7a, b and Supplementary Fig. 4a). Hereafter, we classified spermatocytes harboring at least one wholly synapsed chromosome and at least one chromosome (except for sex chromosomes) synapsed in less than 50% of its length or totally unsynapsed as *aberrant pachytene*, in accordance with previous studies in which similar cells were found in mutants and designated as asynapsed pachytene[54,55].

Relying on this criterion, we barely observed aberrant pachytene spermatocytes in in vivo-grown testes. In explants, we found them infrequently at 34 °C and more abundantly at 37 °C. At 38 °C, we observed aberrant pachytene spermatocytes, but no (fully synapsed) pachytene (Fig. 7a–c and Supplementary Fig. 4a). We then stained MLH1 to visualize the site of crossover formation, which usually occurs in late pachytene after stage IV (Fig. 4a)[56]. In in vivo-grown testes, many pachytene spermatocytes harbored ~22–30 foci as reported previously[31,43]. The others showed zero or intermediate numbers, reflecting those before or during the formation of MLH1 foci, respectively (Fig. 7d–g). In explants, we observed no MLH1 foci in aberrant pachytene at any of the temperatures tested (i.e., 34, 37, and 38 °C), even on the fully synapsed chromosomes (Fig. 7d). This observation suggests that aberrant pachytene spermatocytes do not activate the meiotic program that usually occurs in late pachytene after stage IV; this is in line with the absence of Acr-GFP expression at 38 °C (Fig. 1). In contrast, a fraction of fully synapsed pachytene spermatocytes surviving at 34 °C and 37 °C harbored MLH1 foci, indicating their progression to late pachytene. However, the number of MHL1 foci reduced to ~20–27 at 34 °C, and 17–25 at 37 °C (Fig. 7d–g). Given the 19 autosome pairs and a pair of sex chromosomes, chromosome pair(s) lacking MHL1 foci were often observed in pachytene spermatocytes grown at 37 °C (Fig. 7e). Thus, crossover formation was compromised in late pachytene spermatocytes developed ex vivo, subcritically at 34 °C and critically at 37 °C.

**Involvement of LINE1 transposon is not supported**. Finally, we examined the potential involvement of dysregulated transposon expression, considering the increase in DSBs in explants cultured at 37 °C and 38 °C (Fig. 4b, c). In *Caenorhabditis elegans*, a transient heat shock (at 34 °C for 2 h, compared with a standard culture temperature of 20 °C) elicits dysregulated expression of transposons and an increase in RAD51 foci in spermatocytes[57]. Heat-induced transposon activation also occurs in other systems such as *Arabidopsis*[58]. Furthermore, in mice, mutants deficient in the piRNA pathway exhibit dysregulation of transposons (typically LINE1) and apoptotic death of spermatocytes[59–63].

In in vivo-developed testes, we found LINE1-ORF1 protein expression in spermatocytes in a small fraction of seminiferous tubules (Supplementary Fig. 5). However, in explants cultured at 34 °C, LINE1 expression broadened for unknown reasons, but a temperature-dependent increase in LINE1 expression was not observed (Supplementary Fig. 5). To further investigate the relationship between temperature and LINE1 expression, we used an in vivo setting (i.e., artificial cryptorchidism) considering the low LINE1 expression in the normal condition of the scrotum (at ~34 °C). After surgical translocation of the testis to the abdominal cavity (at ~38 °C), germ cell death, including that of spermatocytes, occurs from day 3 onward[64]. On day 2, therefore, the surviving spermatocytes are likely already impaired and about to die. However, we found that LINE1-ORF1 expression remained limited to a minor fraction of seminiferous tubules on day 2, contrary to the upcoming ubiquitous germ cell death

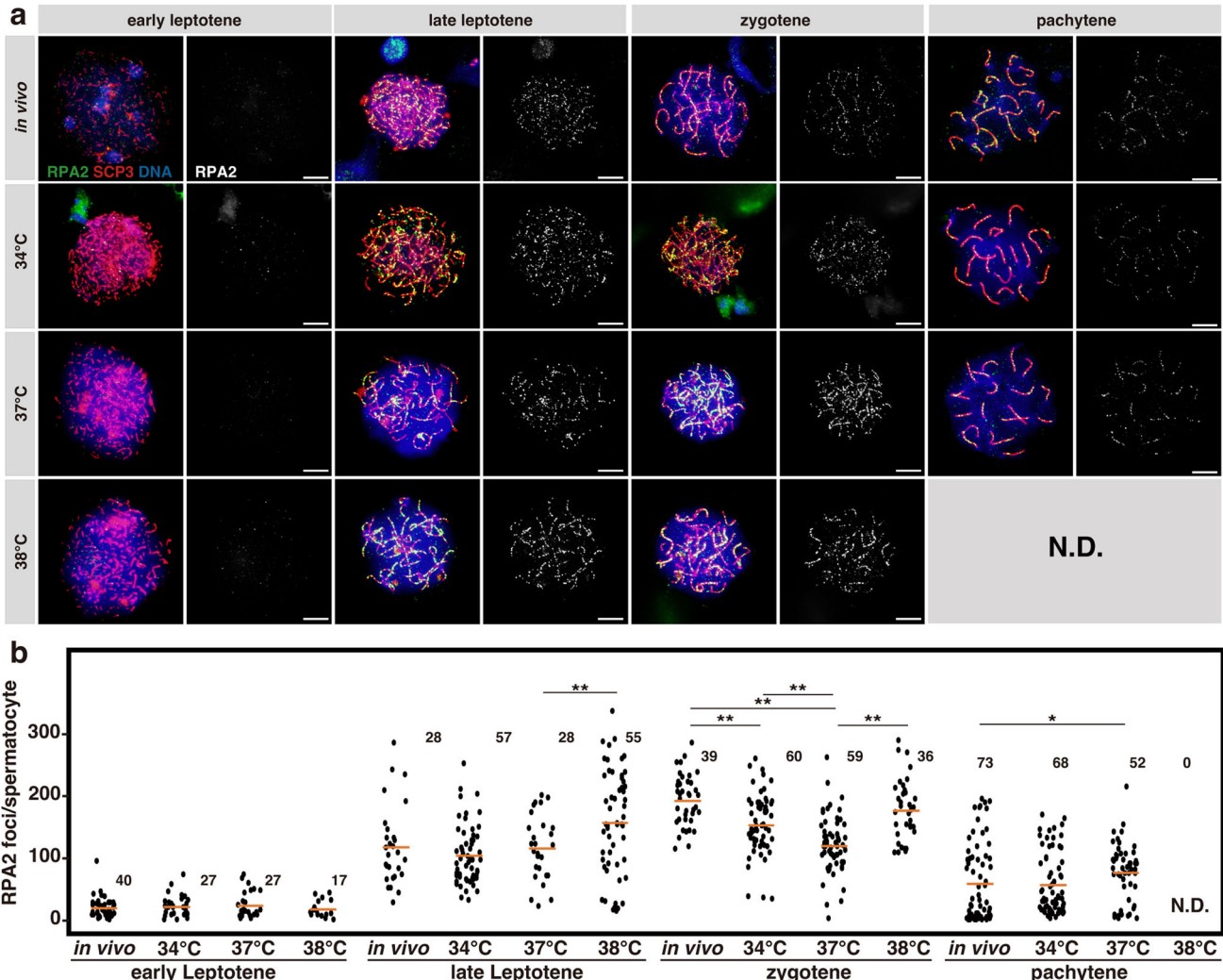

**Fig. 5 Evaluation of RPA foci formation at different temperatures. a** Representative images of chromosomal spreads in meiotic prophase I substages, prepared from the testes of 5-week-old *Acr-GFP* mice and testis explants cultured for 5 weeks at 34, 37, or 38 °C. Samples were stained for RPA2 shown in green, with SCP3 (red) used for staging and DNA (blue). RPA2 signals are shown separately in the right panels. Scale bars, 10 µm. **b** Dot plots showing the number of RPA2 foci associated with SCP3 $^+$ chromosome axes in each spermatocyte of the indicated substage prepared from the in vivo and ex vivo testis samples. Each dot represents an individual spermatocyte with means indicated as horizontal bars. The number of spermatocytes analyzed are indicated above the plots (N.D. not detected). Cells prepared from three testes from different individuals (in vivo) and 8 (34 °C), 8 (37 °C), and 8 (38 °C) explants were pooled and analyzed. Multiple comparisons were conducted using the Kruskal–Wallis test among all values within each category and gave p-values of 0.88, 1.39E-3, 1.13E-11, and 0.02, respectively, for early leptotene, late leptotene, zygotene, and pachytene. The Steel–Dwass test was applied for combinations of two values within each category resulting in the p-values as indicated in the plots (*p < 0.05, **p < 0.01).

(Supplementary Fig. 5a, b). These results indicate that high temperature and LINE1 dysregulation are not parallel but separable events, suggesting that LINE1 does not likely contribute to heat-induced spermatogenesis failure.

## Discussion

In this study, we investigated the temperature sensitivity of mouse spermatogenesis using a testis explant culture that supports the process of spermatogenesis. We found that spermatogenesis was impaired at multiple steps in a temperature-dependent manner. In particular, spermatogenesis did not proceed beyond early spermatocyte, late spermatocyte, and round spermatid at 38, 37, and 36 °C, respectively. Such delicate temperature dependence may explain the varying defects observed in human testes with varicocele[7] and artificial cryptorchidism in mice[9–13,64,65]. Unexpectedly, however, the most severe defect of artificial cryptorchidism, which blocks the transition from undifferentiated to

differentiating spermatogonia[12,13], did not occur ex vivo at any temperature tested. The pathology of cryptorchidism may not be a simple consequence of high temperatures and warrants further investigation.

Further, we investigated meiosis failure observed at 38 °C and 37 °C and found an increase in DSBs and asynapsis. We reason that the *meiotic checkpoint*, also known as the *pachytene* or *synaptic checkpoint*, determines the outcome of such damaged cells. The meiotic checkpoint surveys primary spermatocytes with abundant DSBs and asynapsis, prevents damaged cells from progressing to late pachytene, and eliminates them through apoptosis[66–71]. In mice, a variety of mutants defective in DSB generation (e.g., *Spo11*–/–), single-stranded DNA binding proteins (e.g., *Rpa1* conditional knockout), DSB repair machinery (e.g., *Dmc1*–/–), or their recruitment (e.g., *Meilb2*–/–, *Brme1*–/–, or *Brca2* hypomorph) result in apoptotic cell death at pachytene[72–75]. Further, showing dysregulated transposon

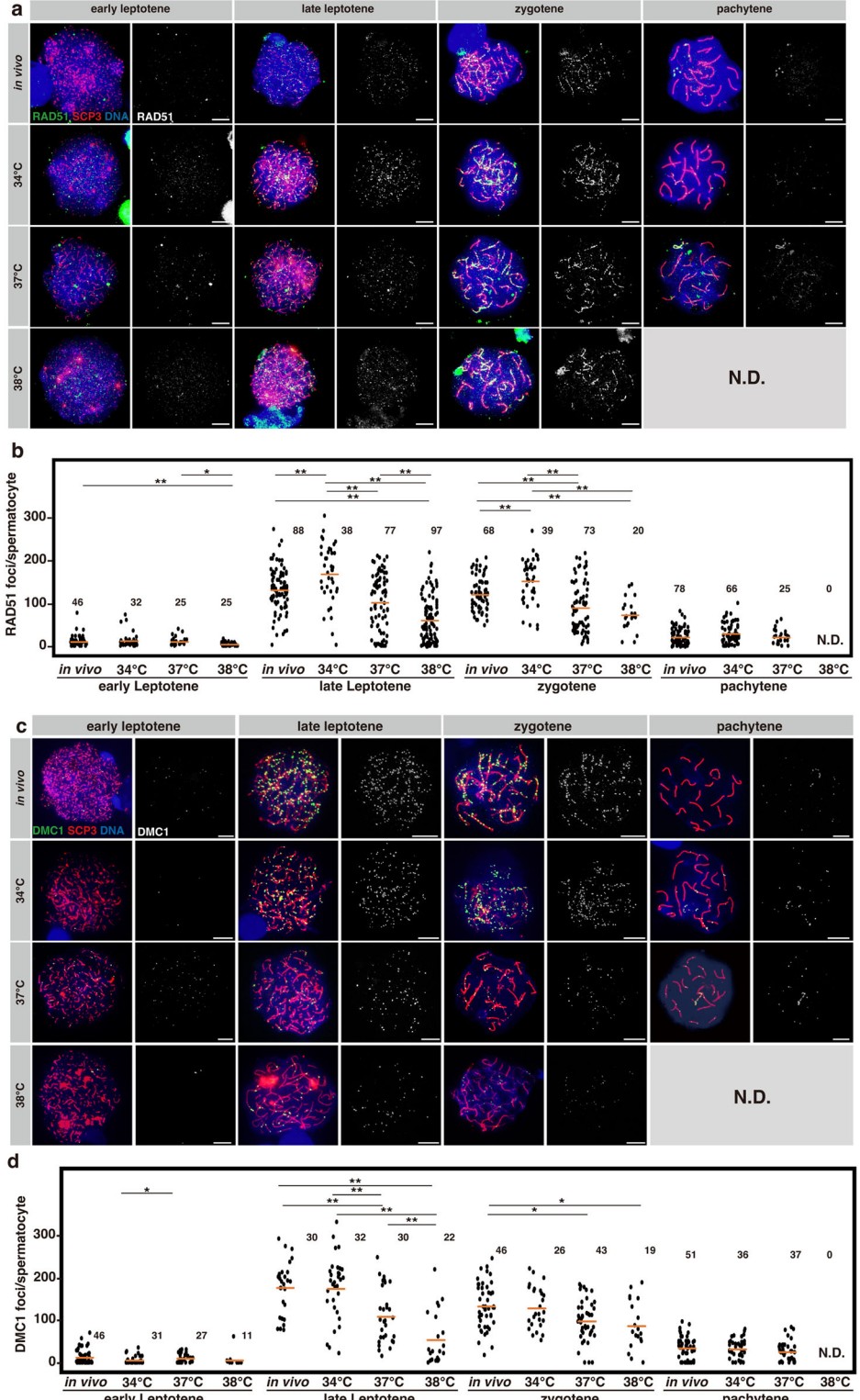

**Fig. 6 Evaluation of repair foci formation at different temperatures. a, c** Representative images of chromosomal spreads in meiotic prophase I substages, prepared from the testes of 5-week-old *Acr-GFP* mice and testis explants cultured for 5 weeks at 34, 37, or 38 °C. Samples were stained for RAD51 (**a**), and DMC1 (**c**) shown in green, with SCP3 (red) used for staging and DNA (blue). RAD51 and DMC1 signals are shown separately in the right panels. Scale bars, 10 µm. **b, d** Dot plots showing the number of RAD51 (**b**), and DMC1 (**d**) foci associated with SCP3⁺ chromosome axes in each spermatocyte of the indicated substage prepared from the in vivo and ex vivo testis samples. Each dot represents an individual spermatocyte with means indicated as horizontal bars. The number of spermatocytes analyzed are indicated above the plots (N.D. not detected). Cells prepared from three, four, and two testes from different individuals (in vivo) and 14 and 4 (34 °C), 15 and 10 (37 °C), and 15 and 4 (38 °C) explants were pooled and analyzed in (**b, d**), respectively. Multiple comparisons were conducted using the Kruskal–Wallis test among all values within each category and gave *p*-values of 0.01, 4.75E-18, 1.68E-9, and 0.11 in (**b**) and 0.02, 9.08E-9, 2.20E-3, and 0.25 in (**d**), respectively, for early leptotene, late leptotene, zygotene, and pachytene. The Steel–Dwass test was applied for combinations of two values within each category resulting in the *p*-values as indicated in the plots (*$p < 0.05$, **$p < 0.01$).

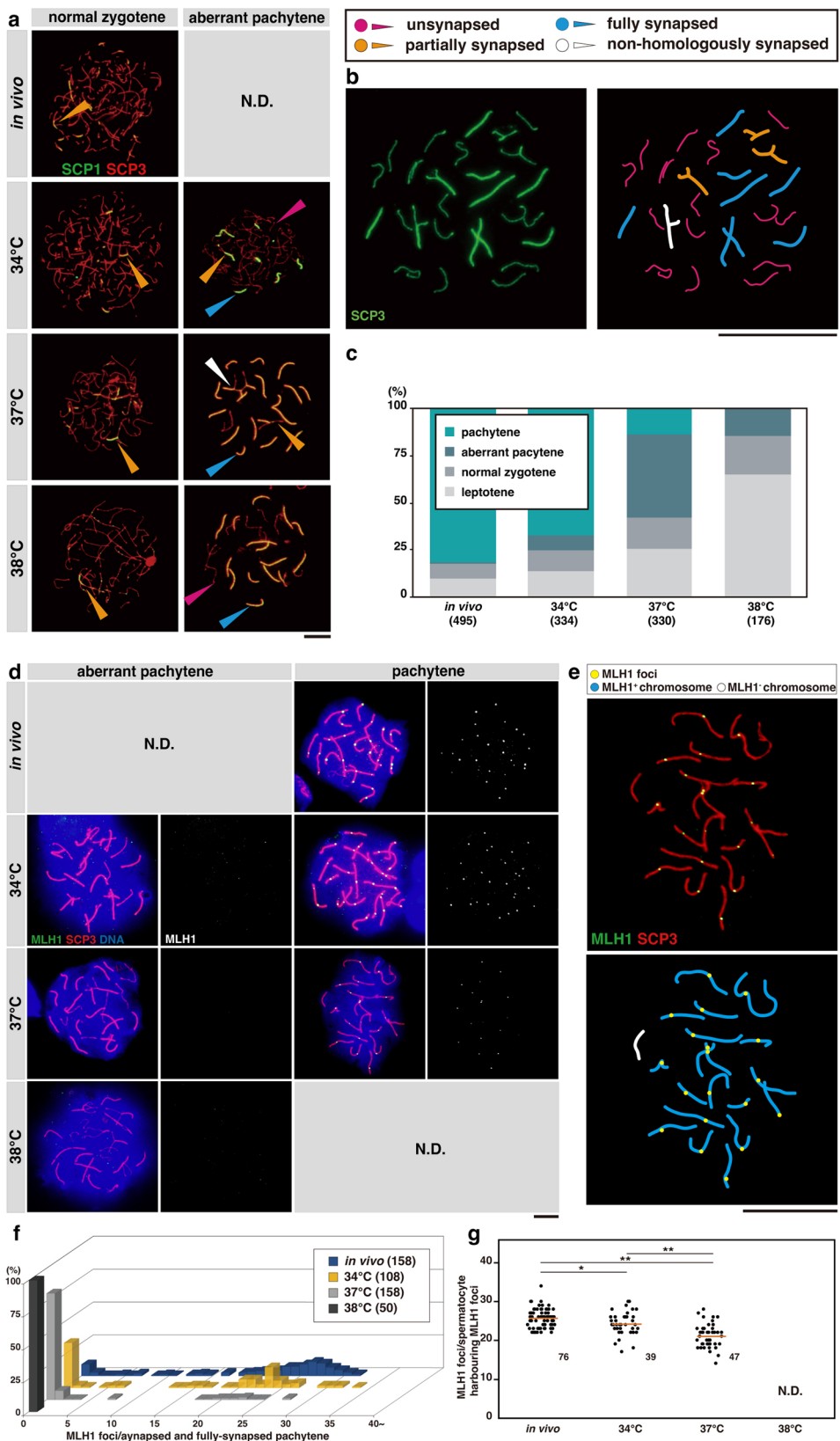

expression associated with increased DSBs, various mutations in the piRNA pathway cause apoptosis at the meiotic checkpoint[59–61].

Figure 8 summarizes the temperature-dependent meiosis failure and the role of meiotic checkpoints that we propose. At 34 °C,

prophase I progresses in a largely normal manner, leading to the production of functional sperm capable of fertilizing eggs[20], although there is a slight increase in DSBs and a small reduction in crossovers. At 38 °C, however, much greater DSBs occur at leptotene and zygotene, at least in part, due to defective DSB

**Fig. 7 Evaluation of the chromosome synapsis and crossover formation at different temperatures. a** Representative images of normal zygotene and aberrant pachytene classified based on the criteria described in the text, observed in chromosome spreads prepared from the testes of 5-week-old *Acr-GFP* mice and explants cultured for 5 weeks at 34, 37, and 38 °C. Samples were stained for SCP1 (green) with SCP3 (red) used for staging. Scale bar, 10 µm. N.D. not detected. Aberrant pachytene is characterized by the co-existence of unsynapsed (red arrowheads), partially synapsed (orange), and fully synapsed (blue) autosomes and, occasionally, non-homologous chromosome pairing (white) within a single nucleus. **b** (left) A typical photomicrograph of aberrant pachytene spermatocytes observed in the chromosomal spread prepared from explants cultured at 37 °C for five weeks stained for SCP3. (right) A traced image of the image on the left indicating unsynapsed (red), partially synapsed (orange), fully synapsed (blue), and non-homologously synapsed (white) chromosomes. Scale bar, 20 µm. **c** Proportion of leptotene, normal zygotene, aberrant pachytene, and pachytene spermatocytes in in vivo-grown testes or explants cultured for five weeks at indicated temperatures. Cells prepared from four testes from different individuals and those from 10 (34 °C), 15 (37 °C), and 10 (38 °C) explants were pooled and analyzed. The number of spermatocyte nuclei analyzed are indicated below. **d** Representative images of chromosome spreads of aberrant pachytene and fully synapsed pachytene spermatocytes, prepared from the testes of 5-week-old *Acr-GFP* mice and explants cultured for five weeks at 34, 37, or 38 °C, stained for MLH1 (green), with SCP3 (red) used for staging and DNA (blue). MLH1 signals are shown separately in the right panels. Scale bar, 10 µm; N.D. not detected. **e** (left) A typical photomicrograph of pachytene spermatocytes observed in the chromosomal spread prepared from explants cultured at 37 °C for 5 weeks, stained for SCP3 and MLH1. (right) A traced image of the image on the left indicating MLH1$^+$ (blue) and MLH1$^-$ (white) chromosomes. Scale bar, 20 µm. **f** Frequency distributions of spermatocytes harboring the indicated number of MLH1 foci. Aberrant pachytene and fully synapsed pachytene spermatocytes found in the indicated in vivo and ex vivo samples were counted. Cells prepared from 3 testes of different individuals and those from 10 (34 °C), 12 (37 °C), and 12 (38 °C) explants were pooled and analyzed. The number of spermatocyte nuclei analyzed are indicated in parentheses. **g** Number of MLH1 foci associated with SCP3$^+$ axes in individual pachytene spermatocytes harboring intense and discrete MLH1 foci, to assess the number of crossovers formed in the surviving cells. Such cells represent the groups in (**f**) with MLH1 foci in numbers comparable with or more than the chromosome pairs (i.e., 20); the remaining cells in (**f**) harbor no MLH1 foci or only slight and faint foci presumably during the assembly process. Horizontal bars indicate the mean values; number of spermatocytes counted are shown beside the plots; N.D. not detected. Cells from 3 testes from different individuals (in vivo) and 10 (34 °C), 12 (37 °C), and 12 (38 °C) explants were pooled and analyzed. $p = 9.53\text{E-}13$ by multiple comparison (Kruskal–Wallis test); *$p < 0.05$, **$p < 0.01$ for indicated pairs (Steel–Dwass test).

repair machinery (i.e., reduced RAD51 and DMC1 foci). Abundant DSBs cause severe asynapsis between homologous chromosomes, leading to *aberrant pachytene* spermatocytes. In these cells, unrepaired DSBs and asynapsis exceed the threshold, activating the meiotic checkpoint for apoptosis. At 37 °C, the damaged cells show a broad range of DSBs and asynapsis, from levels similar to 34 °C to those comparable to 38 °C, reflecting the variable defects in DSB repair foci assembly. Here, cells carrying DSBs and asynapsis above the threshold may result in aberrant pachytene and undergo apoptosis. While, cells whose DSBs and asynapsis are below the threshold complete the chromosome pairing and are licensed to proceed to late pachytene. However, these cells do not survive beyond meiotic divisions, with unrepaired DSBs and asynapsis at levels higher than the normal range compromising the crossover formation critically.

The meiotic checkpoint is an evolutionarily conserved surveillance mechanism that eliminates damaged spermatocytes in various conditions. These include the meiosis block in interspecific hybrids, which leads to hybrid sterility, a vital means of speciation[76]. The current study suggests that the meiotic checkpoint eliminates heat-damaged spermatocytes. This observation echoes the fact that most mammals cool their testes in the scrotum or through other mechanisms[1,3,77,78]. We propose that the meiotic checkpoint serves as a safety device, ensuring that only spermatozoa developed at low temperatures can fertilize eggs. Future investigations are warranted to understand the evolutionary constraint to cool the testes and eliminate damaged germ cells.

The molecular mechanism underlying this delicate temperature sensitivity remains unclear. However, we would hypothesize some mechanisms explaining the meiotic failure at 38 °C and, to a weaker extent, at 37 °C. The findings of this study suggest that the increase in DSBs in leptotene and zygotene is the primary cause of meiotic failure at high temperatures. This appears to be, at least in part, due to the defective assembly of the DSB repair machinery on the DSBs, given the reduced RPA2, RAD51, and DMC1 foci. The potential underlying heat-sensitive processes include DNA resection generating single-stranded DNA, RPA binding to single-stranded DNA, and interaction between these proteins[48]. Alternatively, the amount of RPAs, RAD51, DMC1, or

other proteins can be reduced at high temperatures. Furthermore, the foundation of the repair foci assembly provided by the axis, synaptonemal complex, and aligned chromosomes may be temperature-sensitive, as observed in other organisms, including *Arabidopsis* and *C. elegans*[79,80]. However, these factors also play roles in oogenesis, which occurs at the body core temperature in embryonic ovaries[81] and at 37 °C in vitro as seen in a recently established culture using pluripotent cell-derived PGC-like cells[82,83]. Therefore, male-specific factor(s) must be crucial for temperature sensitivity, for example, sex-specific components of the DSB repair machinery or structures of the synaptonemal complex. Such sex dependence contradicts the possibility that the function of RPAs, RAD51, or DMC1 proteins per se is temperature-sensitive.

Although defects in DSB repair are crucial for heat-induced meiotic failure, additional mechanisms may also be involved. In particular, the generation of DSBs can increase at high temperatures. In this regard, a considerable possibility is the dysregulation of transposons by heat, as recently demonstrated in *C. elegans*[57]. In mice, while LINE1 is unlikely to contribute to the temperature-dependent increase in DSBs, the involvement of other transposons is an open question that warrants further investigation. The assembly of MLH1$^+$ late recombination nodules or crossover formation might also be heat-sensitive. More generally, temperature effects may alter over time between the initial (acute) and late (chronic) phases owing to the *hormesis* effect. Thorough and careful investigations are necessary to fully understand the mechanisms of heat-induced meiosis failure.

By nurturing spermatogenesis at arbitrary temperatures, free from extratesticular factors, explant culture provides an invaluable and otherwise unavailable opportunity to study heat vulnerability in spermatogenesis. At the same time, the current culture supports spermatogenesis less efficiently than in vivo development, even at the optimum temperature (34 °C), which could modify the effect of temperature. Such inefficiency may reflect the subcritical defects observed at 34 °C, including expansion of LINE1 expression, a mild increase in DSBs, and a slight decrease in crossovers. These findings may help further optimize culture conditions and improve the usefulness of the explant culture.

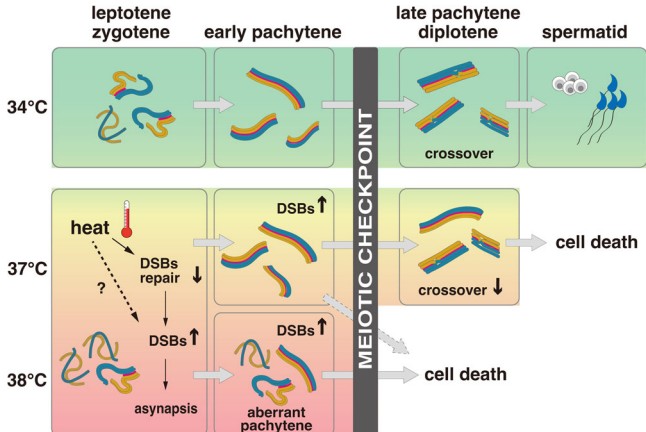

**Fig. 8 Schematic representation of the impact of temperature on mouse meiosis.** At 34 °C, meiosis is completed leading to spermatid formation. At high temperatures (37 °C or 38 °C), DNA double-strand breaks (DBSs) increased, at least in part, due to defective DSB repair, leading to increased asynapsis. Heat may also increase the DSB genesis. The degree of impairment varies between cells, with severely affected cells falling into a state of aberrant pachytene and less-affected cells completing chromosome pairing. The meiotic checkpoint then surveys the DSBs and asynapsis in these cells. Cells with DSBs and asynapsis above the threshold (including aberrant pachytene and probably some fully synapsed early pachytene) are eliminated through apoptotic cell death; those below the threshold survive to late pachytene. All cells are eliminated at the meiotic checkpoint at 38 °C, whereas some survived to late pachytene at 37 °C. However, late pachytene spermatocytes surviving at 37 °C retain abnormally abundant DSBs and asynapsis, leading to compromised crossover formation and cell death without completing meiotic divisions.

In conclusion, we hope that ex vivo culture and the findings of this study will help to better understand the mechanisms of temperature sensitivity in mammalian spermatogenesis. In addition to developing an infertility treatment strategy, the results would also provide insights into finding the answer to the fundamental question of why does mammalian spermatogenesis proceeds at low temperatures.

## Methods

**Mice.** *Acr*-GFP transgenic mice, designated formally as B6;B6C3-Tg(Acro3-EGFP) 01Osb (RIKEN BRC, #RBRC00886)[21,22], with a genetic background of C57BL/6 J (Japan SLC and Japan CLEA), were used for histological and immunohistological analysis and ex vivo organ culture experiments. C57BL/6 J mice were used for temperature measurements and histological and immunohistological analysis of testes. All mouse experiments were conducted with approval from the Institutional Animal Care and Use Committee of the National Institutes of National Sciences.

**Artificial cryptorchidism.** Eight-week-old C57BL/6 J mice were subjected to operations of artificial cryptorchidism following a rigorous method wherein the testes were wrapped by the associated fat pad and positioned deep in the abdomen[13].

**Measurement of the scrotal and body core temperatures in mice.** IPPT-300 transponders (BioMedic Data Systems) were implanted into the scrotal sac and peritoneal cavity (under the liver) of wild-type C57BL6/J mice. One week after surgery, the temperature at each position was measured separately using a DAS-7006S scanner (BioMedic Data Systems) every 2 h for 24 h. Ambient conditions were maintained under a 12 h light/12 h dark at room temperature (25 ± 2 °C).

**Ex vivo culture of mouse testis explants under different temperatures.** Ex vivo culture was performed as previously described[20,84,85] with minor modifications. Briefly, 1.5% (wt/vol in PBS) agarose gels (Dojindo Laboratory, Kumamoto, Japan) were prepared and submerged in the culture medium αMEM/10% KnockOut Serum Replacement/1% Penicillin–Streptomycin for >24 h, repeatedly 2–3 times. Whole testes dissected from euthanized neonatal *Acr*-GFP mice (4 days postpartum) were removed from the tunica albuginea and, without cutting into

multiple pieces, gently positioned on the agarose gel in a 12-well culture plate with 500 μL of the above-described medium. The medium was changed once a week for 5 weeks. The culture incubators (APC-30DR; Astec) were supplied with 5% $CO_2$ in the air and maintained at specific temperatures (ranging from 30 to 40 °C), controlled with an accuracy of ±0.2 °C.

The grading of spermatogenesis progression based on Acr-GFP expression was performed visually at the time of weekly medium change, using an IX73 inverted fluorescence microscope (Olympus). Grading was performed unblinded as the observation had to be completed in a short period to minimize the temperature fluctuation.

**Histological analysis.** Testes removed from euthanized mice or cultured testis explants were fixed overnight in Bouin's solution or 4% PFA in PBS at 4 °C, embedded in paraffin, sectioned, and mounted on a glass slide. According to standard protocols, sections were stained with hematoxylin and eosin (HE) or hematoxylin and periodic acid-Schiff (H-PAS) and mounted for microscopy.

**Immunofluorescence on sections.** Immunofluorescence (IF) was performed on paraffin sections (7 μm thick) for most cases. For double staining of LINE1-ORF1 and SCP3 (Supplementary Fig. 5), samples were fixed overnight in 4% PFA in PBS at 4 °C, embedded in Tissue-Tek O.C.T. compound (Sakura Finetek), cryosectioned (8–10 μm thick), and attached to MAS-GP type A-coated slides (Matsunami). Immunostaining was performed following the protocol described previously with minor modification[86]. Briefly, the sections were incubated with Blocking One Histo (Nacalai Tesque) solution for 1 h at room temperature, followed by incubation with primary antibodies (described below) in 4% donkey serum/Can Get Signal Solution 1 (Toyobo) overnight at 4 °C. After the sections were washed with 0.04% Tween 20 in PBS, they were incubated with appropriate Alexa Flour-conjugated secondary antibodies (Thermo Fisher Scientific and Jackson ImmunoResearch) in 4% donkey serum/Can Get Signal Solution 2 containing Hoechst 33342 for 2 h at room temperature. After the slides were washed with the secondary antibody solution, they were mounted using Fluoro-KEEPER Antifade Reagent (Nacalai) for fluorescence microscopy.

The following primary antibodies were used at the indicated dilutions: anti-GFRα1 goat polyclonal antibody (R&D Systems, #AF560, 1:1000); anti-KIT goat monoclonal antibody (R&D Systems, #AF1356, 1:1000), anti-mouse Vasa homolog (MVH) rabbit polyclonal antibody (Abcam, #ab13840, 1:500); anti-SCP3 mouse monoclonal antibody (Abcam, #ab97672, 1:1000), anti-GATA4 rabbit polyclonal antibody (Thermo, #PA1-102, 1:500); anti-cleaved Caspase-3 rabbit polyclonal antibody (Cell Signaling, #9661, 1: 500); and anti-LINE1-ORF1 rabbit polyclonal antibody (a generous gift from O'Carroll D, The University of Edinburgh, Edinburgh, UK;[87], 1:5000) (Supplementary Table 2).

**Nuclear spread of spermatocytes.** The nuclear spread of spermatocytes from in vivo-developed testes and cultured testis explants was prepared using the dry-down method as described previously[51,88], with modifications. Briefly, the tissues were minced with scissors and then dissociated into single cells by vigorous pipetting and filtration through a nylon mesh with 70 μm pores (Becton Dickinson). Cells were then suspended in PBS for hypotonic treatment by mixing with an equal volume of 30 mM Tris, 50 mM sucrose, 17 mM trisodium citrate dihydrate, 5 mM EDTA, and 0.5 mM dithiothreitol. Cells were washed and resuspended in PBS, mixed twice with 100 mM sucrose solution (pH 8.2), and then with three volumes of 1% PFA/0.1% (v/v) Triton X-100 in 10 mM sodium borate, before being dropped onto a glass slide. The slides were then incubated in a humidified chamber for 1 h at room temperature, and air-dried and then frozen at −80 °C before IF. To obtain sufficient amounts of starting materials to avoid the massive loss because of their small size during these procedures, several (e.g., 4–8) ex vivo explants were pooled.

For IF, the specimens were first blocked with 10% or 2% fetal bovine serum in 0.05% Triton X-100 in PBS (blocking buffer) for 1 h at room temperature and then with primary antibodies (listed below) in blocking buffer overnight at 4 °C. The specimens were then washed with 0.05% Triton X-100 in PBS, and incubated with Alexa Flour-conjugated secondary antibodies diluted in blocking buffer containing Hoechst 33342 for 2 h at room temperature before they were washed and mounted for fluorescence microscopy.

The primary antibodies used are as follows: anti-DMC1 rabbit polyclonal antibody (Santa Cruz, #sc-22768, 1:1000), anti-SYCP1 polyclonal antibody (NOVUS Biologicals, #NB300-229, 1:500), anti-SYCP3 rabbit polyclonal antibody (Abcam, #ab15093, 1:500), anti-SCP3 mouse monoclonal antibody (Abcam, #ab97672, 1:1000), anti-phospho-H2A.X (Ser139; Millipore, #07-164, 1:500), anti-RPA2 (Cell Signaling Technologies, #2208, 1:80), anti-RAD51(Bio Academia, #70-002, 1:100), anti-MLH1 (BD Biosciences, #551092, 1:100), and anti-HORMAD1 (Proteintech, #13917-1-AP, 1:1000) (Supplementary Table 2).

**Image acquisition and processing.** The sectioned specimens on glass slides (HE, PAS-hematoxylin, or immunostained samples) were observed and photographed using a BX51 upright fluorescence microscope equipped with a DP72 CCD camera operated using a cellSens imaging software (Olympus). Observations and signal quantification of γH2AX staining (Fig. 4b, c) were performed using a Leica SP8

confocal microscope. For explants, GFP fluorescence was observed and photographed using an IX73 inverted fluorescence microscope equipped with a DP72 CCD camera operated using cellSens (Olympus). Photoshop (Adobe) was used to adjust the signal intensity uniformly over each image, without losing the original information. An illustrator (Adobe) was used to create figure panels.

**Statistics and reproducibility**. Figure 1a: Both scrotal and abdominal temperatures were measured in 14 wild-type mice at 12 timepoints at 2 h intervals (16:00 to 14:00 on the next day). The scrotal and abdominal temperatures were averaged over the 12 timepoints for each mouse. Such averaged values (per individual) were further averaged among 14 mice, shown with SD. Statistical evaluation was performed using matched Student's $t$-test ($p = 4.6$ E-11).

Figure 1e: The Acr-GFP grade (0–4) of testicular explants, determined visually under a fluorescence microscope once per week during the culture period of five weeks at 34 or 38 °C, is summarized in component bar charts. 99 and 51 explants were analyzed for 34 and 38 °C cultures, respectively.

Figure 2a: The Acr-GFP grade (0–4) of testicular explants determined in the same manner described in Fig. 1e at a range of different culture temperatures is summarized in component bar charts. The number of explants were 6, 4, 99, 15, 9, 41, 51, 11, and 11 for 30, 32, 34, 35, 36, 37, 38, 39, and 40 °C, respectively. Data for cultures at 34 °C and 36 °C are identical to those shown in Fig. 1e.

Figure 3b: Compositions of prophase I spermatocytes in different substages (leptotene, zygotene, pachytene, or diplotene). The total number of spermatocytes counted was 420, 549, 496, and 456 for the in vivo-developed testes (2 testes from different individuals) and testis explants cultured for five weeks at 34 °C (6 explants), 37 °C (7 explants), and 38 °C (7 explants), respectively. The number of spermatocytes in the leptotene stage was 108, 159, 211, and 253, those in the zygotene stage were 49, 105, 206, and 201, those in the pachytene stage were 200, 259, 56, and 2, and those in the diplotene stage were 63, 26, 23, and 0, for the in vivo-developed testes and testis explants cultured at 34, 37, and 38 °C, respectively.

Figure 4c: To quantify the $\gamma$H2AX signal, we acquired immunofluorescence images of meiotic chromosome spread prepared from in vivo and ex vivo testicular cells stained for $\gamma$H2AX and SCP3 (as shown in Fig. 4b), using the TCS SP8 confocal system (Leica). The $\gamma$H2AX signal intensity of individual spermatocytes was quantified as the sum of pixel values within a nucleus using the Fiji software[89], with cells classified visually in the leptotene, zygotene, pachytene, or diplotene stage based on the SCP3 staining pattern. Four testes from different individuals were pooled for the in vivo sample, whereas cells prepared from 14 (34 °C), 15 (37 °C), and 15 (38 °C) explants were pooled for each ex vivo sample to prepare the nuclear spread specimens. As indicated in the panel, the number of spermatocytes analyzed to be in the leptotene stage was 73, 59, 79, and 98, those in the zygotene stage were 54, 62, 76, and 49, those in the pachytene stage were 89, 85, 71, and 0, and those in the diplotene stage were 68, 47, 45, and 0 for the in vivo and ex vivo explants at 34, 37, and 38 °C, respectively. In the dot plots, the values of the $\gamma$H2AX signal are shown in the AU. Multiple comparisons were conducted using the Kruskal–Wallis test among all values within each category, providing $p$-values of 2.31E-34, 1.13E-19, 2.53E-19, and 3.76E-5 for leptotene, zygotene, pachytene, and diplotene, respectively, and using the Steel–Dwass test for combinations of two values within each category, resulting in $p$-values as indicated in the plots (*$p < 0.05$, **$p < 0.01$).

Figures 5b, 6b, 6d: To quantify the RPA2 (5b), RAD51 (6b), and DMC1 (6d) foci, we prepared chromosome spread specimens from the testes of 5-week-old Acr-GFP mice and testis explants cultured for five weeks at 34, 37, or 38 °C, followed by immunostaining for RPA2 (5b), RAD51 (6b), or DMC1 (6d) and co-staining with SCP3. Photomicrographs were taken using a BX51 upright fluorescence microscope equipped with a DP72 CCD camera operated using cellSens imaging software (Olympus). Dots of RPA2 (5b), RAD51 (6b), or DMC1 (6d) associated with SCP3$^+$ axes were counted manually, and each spermatocyte nucleus was classified to be in the early leptotene, late leptotene, zygotene, or pachytene stage. Cells prepared from three, four, and two testes from different individuals (in vivo) as well as 8, 14, and 4 (34 °C), 8, 15, and 10 (37 °C), and 8, 15, and 4 (38 °C) explants were pooled and analyzed in (5b), (6b), and (6d), respectively. The number of spermatocytes examined is indicated in the panels (N.D. not detected). Multiple comparisons were conducted using the Kruskal–Wallis test among all values within each category, with $p$-values of 0.88, 1.39E-3, 1.13E-11, and 0.02 in (5b), 0.01, 4.75E-18, 1.68E-9, and 0.11 in (6b), and 0.02, 9.08E-9, 2.20E-3, and 0.25 in (6d), respectively, for early leptotene, late leptotene, zygotene, and pachytene stage. The Steel–Dwass test was applied for combinations of two values within each category, resulting in $p$-values as indicated in the plots (*$p < 0.05$, **$p < 0.01$).

Figure 7c: Proportions of leptotene, normal zygotene, aberrant pachytene, and (fully synapsed) pachytene spermatocytes contained in testes developed in vivo or explants cultured for five weeks at indicated temperatures were classified visually based on chromosome spreads stained for SCP3 and SCP1. Leptotene spermatocytes were identified following authentic criteria, that is, the presence of SPC3$^+$ chromosome axes and the absence of SCP3$^+$/SCP1$^+$ synapsed regions. *Normal zygotene* spermatocytes were those classified as zygotene spermatocytes (i.e., the presence of both SCP3$^+$/SCP1$^-$ unsynapsed region and SCP3$^+$/SCP1$^+$ synapsed region) but not as *aberrant pachytene* described below. *Aberrant pachytene* spermatocytes were in an unusual state of zygotene (though confusing literally), harboring one or more fully synapsed chromosomes and one or more autosome pairs that were unsynapsed or were partly

synapsed in less than 50% of its length. Pachytene spermatocytes were defined authentically when all chromosomes had fully synapsed except for the XY body. The total number of spermatocytes counted was 495, 334, 330, and 176 for in vivo-developed testes (prepared from four testes of different individuals) and testis explants cultured for five weeks at 34 °C (10 explants), 37 °C (15 explants), and 38 °C (10 explants), respectively. The resultant counts of leptotene, normal zygotene, aberrant pachytene, and pachytene spermatocytes were 49, 36, 3, and 407 in in vivo-developed testes and 45, 37, 27, and 225 in explants at 34 °C, 113, 44, 134, and 39 in explants at 37 °C, and 113, 36, 27, and 0 in explants at 38 °C, respectively.

Figure 7f: The MLH1 foci associated with SCP3$^+$ axes were counted manually in spermatocytes harboring at least one fully synapsed chromosome [including aberrant pachytene and (fully synapsed) pachytene spermatocytes], on chromosome spread stained for MLH1 and SCP3. Cells prepared from three testes of different individuals and those from 10 (34 °C), 12 (37 °C), and 12 (38 °C) explants were pooled and analyzed. The total number of spermatocytes counted were 158, 108, 158, and 50, respectively.

Figure 7g: The number of MLH1 foci was counted in the same manner as described above for Fig. 7f. However, only pachytene spermatocytes harboring intense and discrete MLH1 foci were counted to assess the crossover formation in the surviving cells. Three testes of different individuals and those from 10 (34 °C), 12 (37 °C), and 12 (38 °C) explants were pooled to prepare the chromosome spread; 76, 39, 47, and 0 spermatocytes were analyzed for in vivo samples and explants at 34, 37, and 38 °C, respectively. The number of MLH1 foci per nucleus is summarized in dot plots, with the mean values shown by horizontal bars. Multiple comparisons were conducted using Kruskal–Wallis test, providing the $p$-values of 9.53E-13, and Steel–Dwass test for combinations of two values, resulting in the $p$-values as indicated in the panel.

Supplementary Fig. 1a: The scrotal and abdominal temperatures in WT (C57BL/6) mice were measured using transplanted thermometers in the scrotum and abdominal cavity (under the liver) every 2 h over a day (from 16:00 to 14:00 on the next day). Data from 14 individuals were collected to evaluate daily temperature changes. Data points are shown as average ± SD. *$p < 0.01$ between scrotal and abdominal temperatures based on statistical evaluation using a matched Student's $t$-test.

Supplementary Fig. 5b: The fraction of seminiferous tubules harboring LINE1-ORF1$^+$ spermatocytes of those with SCP3$^+$ spermatocytes was counted in the in vivo-developed 5-week-old mouse testes, artificial cryptorchid testes (2 days after surgery), and testis explants cultured for five weeks at 34, 36, 37, and 38 °C. Averaged percentage ± SEM calculated from five in vivo testes, five artificial cryptorchid testes, and 5 (34 °C), 5 (36 °C), 6 (37 °C), and 4 (38 °C) explants, respectively. Student's $t$-test was used to determine $p$-values as indicated (*$p < 0.05$, **$p < 0.01$).

**Software**. All calculations and statistical analyzes were performed using Microsoft Excel 2019, KaleidaGraph, Bell Curve for Excel, and R (Supplementary Table 3).

**Reporting summary**. Further information on research design is available in the Nature Research Reporting Summary linked to this article.

## Data availability

Source data behind the graphs in this article can be found in the Supplementary Data 1. Unprocessed images of this article are available at Figshare (https://doi.org/10.6084/m9.figshare.19493555)[90]. Further information and requests for resources and reagents should be directed to and will be fulfilled by lead contact, S.Y. (shosei@nibb.ac.jp).

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

## Acknowledgements

We thank Katsuhiko Hayashi and Yuki Takada for their help with nuclear chromosomal spread preparation, Donal O'Carroll for critical reading of the manuscript and providing the anti-LINE orf1 antibody, and Benjamin D Simons for critical reading of the manuscript. We also thank the Yoshida laboratory members for their fruitful discussions and encouragement. We acknowledge the support of the NIBB Core Research Facilities and Model Animal Research Facility, NIBB Bioresource Center for Animal Care. This work was supported in part by a Grant-in-Aid for Scientific Research (KAKENHI; JP18H05551 to S.Y., JP16H01398 to Yo.N., JP18H05546 to T.O., and JP16H01257, JP19H05245, and JP19H05743 to K.I.), and AMED-CREST (JP18gm0910011h0002 to S.Y.). K.H. and Yu.N. received a Research Fellowship for Young Scientists from JSPS (JP17J08368 and JP15J07680, respectively). To.S. was a fellow of the Takada Science Foundation.

## Author contributions

K.H., Y.No., Y.Na., K.I., T.O., and S.Y. designed and conceptualized the research framework; K.H., Y.No., Ta.S., and T.O. prepared the testicular ex vivo explant culture; K.H. and K.I. analyzed meiosis progression; K.H. and To.S. performed the quantitative measurement of chromosome spread; K.H. and S.Y. drafted the manuscript based on input from all authors.

## Competing interests

The authors declare no competing interests.
