## [Peer Review File · Communications Biology]

Reviewers' comments:

Reviewer #1 (Remarks to the Author):

In this study, the authors investigate the effects of temperature in germ cell differentiation and meiotic recombination in mouse testis explants (a system previously developed by the same lab). In this context, 34°C seemed to be the temperature most compatible with spermatogenesis and spermiogenesis, while lower (30°C, 32°C) and higher (35-40°C) temperatures blocked germ cell differentiation at different stages with both temperature extremities severely resulting in total germ cell depletion. The authors then focused in further exploring the underlying mechanisms for the observed meiotic prophase I blockage at 37°C and 38°C. They showed that at these temperatures, late spermatocytes (at pachytene and diplotene) are rare or absent. They then related this to an increase in the staining of apoptotic markers at 38°C comparatively to 34°C. Finally, the authors analyzed homolog synapsis and DMC1 foci numbers, as means to investigate meiotic recombination efficacy at 37°C and 38°C comparatively with in vivo- and ex vivo-grown (at 34°C) testes. The authors found synapsis defects in zygotene spermatocytes and a significant reduction in DMC1 foci numbers in late leptotene and zygotene spermatocytes at 37°C and 38°C. The authors then conclude that heat impairs meiotic recombination and synapsis of homologs which leads to a block in zygotene-pachytene transition and meiotic checkpoint-induced apoptosis.

Although the study is interesting, in my opinion the data shown is still quite preliminary for the claims that the authors want to make. In particular, the analyses of the impact of temperature in spermatocytes at the mechanistic level are superficial and need to be further explored. My specific comments are as follows:

Major comments:

1. The culture of testis explants at 34°C seems to already lead to a significant decrease in cell density in the seminiferous tubules (Fig. 1d,e), suggesting that germ cells are already under some stress in these conditions. This could be further exacerbated by combining with the temperature shift in the explants. Thus, in my opinion the effects that the authors see with temperature shift should be taken with a grain of salt as a synergistic effect between ex vivo testis culture and temperature shift cannot be completely excluded based on the experiments of this study. To confidently support all the conclusions stated by the authors, the comparison with in vivo testes would be the most appropriate, although I understand that this would be unfeasible to do in a short time frame. Therefore, this aspect should be at least discussed in the manuscript.

2. It is unclear how the authors quantified the Acr-GFP signal in the testis explants (could not find any related information in the methods section). In this regard, the GFP signal in images of testis explants grown at 34°C (Extended Data Figure 1d) is not as uniformly distributed and no example fully recapitulates the signal of grade 4 in Extended Data Figure 1c, even though the majority of explants are grade 4 based on the quantifications (Figure 1b). Some regions of the 34°C-grown testis explants have no GFP signal and it is unclear whether this is because there are no tubules. I would strongly suggest the authors to include in Extended Data Figure 1c images showing another marker for tubules and clarify how the number of tubules was taken into consideration when quantifying the Acr-GFP signal.

3. Are the developmental stages of the seminiferous tubules shown in Fig 1d,e, Fig 2b,c, and Extended Data Figures 3 and 4 equivalent? This is important because different developmental stages of the seminiferous tubules have different proportions of distinct germ cell types which could affect the interpretation of these analyses.

4. I am not sure if I agree completely with the conclusions that the authors state in lines 137-141. The results in Figures 1 and 2 show that there is a positive correlation between early defects in germ cell differentiation and temperature increase. This can also be interpreted as a progressive aggravation of the same defects at early stages of germ cell development (for instance in homolog synapsis or DSB repair) such that a threshold is reached at higher temperatures resulting in a complete blockage of germ cell progression. In this respect many studies support a threshold in the number of recombination events needed to support successful chromosome pairing, synapsis,

and segregation in mice (Kauppi et al., 2013; Faieta et al., 2016; Faisal et al., 2016). Furthermore, there is evidence supporting a threshold for unrepaired DSBs to activate the pachytene checkpoint (Pacheco et al., 2015). Supporting this view, the data in Figure3b shows a progressive increase in the percentage of spermatocytes at early prophase and a concomitant decrease of late spermatocytes with the increase of temperature. Thus, I think these points should be taken into consideration by the authors and further discussed in the paper.

5. Related to the point 4, the authors should further comment on the fact that the decrease in the percentage of spermatocytes at pachytene and diplotene stages is accompanied by an increase in spermatocytes at early stages.

6. Given that the authors find synapsis defects in zygotene- and pachytene-like spermatocytes, I strongly recommend the authors to quantify the proportion of these abnormal cells found in each condition and differentiate these in the graphs of Figure3b. Also, please clarify what criteria are used to stage the cells.

7. The authors only use one marker for transposons. The use of a second marker would strengthen the results. The authors should also comment on the increased levels of transposon activity of 34°C-grown explants compared to in vivo grown testis.

8. I recommend that more pachytene cells are analyzed for the 37°C condition in Figure4c because they seem to have significantly less DMC1 foci as shown in Extended Data Figure5 (here no DMC1 foci is seen which is quite different from the in vivo or 34°C conditions).

9. In lines 259-261, the authors state that "This mechanism, discovered and analysed intensively using genetic mutants, plays a role as a guardian of species in inhibiting gametogenesis in interspecific hybrids (a phenomenon known as hybrid sterility)". I am not sure of what the authors mean by this. Extensive synapsis defects is thought to be the underlying cause of infertility in hybrid mice (Bhattacharyya et al., 2013). Moreover, several published studies have shown that the infertility seen in some mouse hybrid strains is caused by incompatibility between the Prdm9 allele of one strain and the genome of the other, such that PRDM9 asymmetric DNA binding affects the efficacy of homolog synapsis (Davies et al., 2016; Mihola et al., 2009; Flachs et al., 2012, Balcova et al., 2016). I recommend that the authors clarify this part and comment on their results in the light of these evidence.

10. The conclusion that the authors make regarding defects in meiotic recombination repair by temperature shift merely based on the decrease in DMC1 foci numbers is faulty. A decrease in DMC1 foci could also result from a decrease in the number of DSBs (as γ H2AX signal was not quantified it is unclear whether this is the case), or even in a decrease in DMC1 protein levels. Thus, I strongly recommend the authors to quantify γ H2AX signal in spermatocytes in the different conditions, as well as the foci number of other DSB markers such as RPA or MEIOB, and RAD51 in these cells. If feasible, western blotting of DMC1 and/or RAD51 to analyse their protein levels should be done as high temperatures could destabilise these proteins. It would also be interesting to analyze the foci of downstream repair factors, such as MSH4 or MLH1, to further understand the impact of temperature in the resolution of non-crossovers/crossovers, and what proportion of the few 37°C-grown pachytene spermatocytes (Figure3b) efficiently repair DSBs and form normal numbers of crossovers.

11. Related to point 11, it seems that increased levels of γ H2AX are present at late spermatocytes grown at 37°C and 38°C. Thus, providing the γ H2AX quantification in these cells will also clarify this which may further support defects in DSB repair.

Minor comments:

1. In the sentence "At 36°C, although similarly intense Acr-GFP signal was observed, elongating spermatids were absent, and a few round spermatids were observed as the most advanced cell type" (lines 125-127) it is not clear to what condition is it comparing to.

2. The use of the terms "synapse formation" (line 151) or "synapse disassembly" (line 153) do not sound right. I would recommend to change it to, for example, "establishing synapsis" and "synaptonemal complex disassembly".

3. "Leptotene spermatocytes..." (line 208) should be late leptotene spermatocytes.

4. The term "homology-dependent mismatch repair" (lines 193) is confusing. Please rephrase it.

5. The way the authors describe the activation of the meiotic checkpoint, best known as the pachytene checkpoint, sounds as they are providing novel findings. It is already well established in the field that there is a mid-pachytene checkpoint that is activated in response to both asynapsis and DSB repair defects. I would recommend to rephrase this section and refer to appropriate studies in literature.

6. The authors often refer to the elimination of spermatocytes on zygotene-pachytene transition, however this is not completely correct. Although defective spermatocytes have incomplete synapsis, they are in a pachytene-like state and they are eliminated by the pachytene checkpoint, as is the case of other mutant spermatocytes (e.g., Pittman et al., 1998; Yoshida et al., 1998; Hamer et al., 2008).

Reviewer #2 (Remarks to the Author):

Spermatogenesis is very temperature sensitive and heat exposure can result in male infertility. How temperature specifically affects the development of sperm is an emerging field. This manuscript used an *ex vivo* mouse testes explant preparation to assess the deleterious effects of chronic heat exposure on spermatogenesis, allowing for the parsing out of heat-sensitive mechanisms specific to the spermatogenic/meiotic program in the absence of potentially confounding physiological inputs. Additionally, the use of chromosome spreads allowed for a more thorough evaluation of the effects of heat on specific meiotic stages. The authors present data that support their claim that relatively small temperature elevations can impair specific stages of spermatogenesis, likely via stage-specific mechanisms. The authors clearly outlined the clinical relevance of this work and compared their specific results back to clinical histological findings to better dissect the mechanisms underlying specific health conditions (e.g. cryptorchidism vs varicoceles).

General comments/concerns

1.) The synaptonemal complex is well documented in multiple systems to be temperature sensitive (reviewed in Morgan CH et al., *Phil Trans R Soc B* 2017; Bilgir et al., *G3* 2013; Loidl *Chromosoma* 1989). Although the SC components SCP1 and SCP3 are traditionally used markers to differentiate the meiotic prophase I stages of unsynchronized spermatocytes, SCP1/3 are also likely to be temperature sensitive, making them unreliable markers for staging of spermatocytes. Using only SCP1/3 to stage spermatocytes weakens the authors' conclusion that chronic heat treatment of spermatocytes causes arrest by zygotene. The authors should use alternative markers for staging their spermatocytes such as MLH1, which loads in pachytene, or perform FISH to assess pairing.

2.) Multiple studies have demonstrated DMC1 only loads to a subset of DSBs in mouse spermatocytes and that RAD51 will load to DSs not destined for homologous recombination in spermatocytes. Based on unquantified H2AX staining and quantified DMC1 staining, the authors conclude that DSB repair is impaired (line 235). This data implies that homologous recombination may be affected (and not that DSB repair in general is impaired). To make this conclusion, the authors need to do the following: 1) quantify H2AX staining; 2) assess/quantify RAD51 staining; and, 3) assess/quantify MLH1 staining (which will determine if homologous recombination is indeed impaired – because MLH1 marks the downstream crossover site that forms in pachytene).

3.) Figure 4C uses a Mann-Whitney U test (non-parametric) to compare 2 sample sets. Performing several Mann-Whitney tests, without a correction, can give spurious results. To compare 2 or more groups, it is more appropriate to run Kruskal-Wallis ANOVA which has the same assumptions as M-W test.

4.) There are differences in the short-term and long-term responses to either chronic or acute heat stress. Specifically, chronic heat stress may elicit a hormesis effect (or biphasic response), therefore an effect may only occur during the initial exposure but then the system adjusts with time (see next comment regarding discussion of transposons). The manuscript addresses the effects on chronic heat stress on spermatogenesis after several days/weeks of heat treatment. The authors should clearly indicate and discuss their results/conclusion in the context of chronic heat stress effects and the effects they may be missing due to hormesis.

5.) Lines 179-187: Using LINE1 alone, the authors conclude that transposons are not activated upon chronic exposure to heat. Several models have recently demonstrated that heat exposure can activate specific classes (or subsets) of transposons and not others (Rusa et al., PNAS 2020; Kurhanewicz et al., Current Biology 2020). Further, transposons could be activated at the onset of heat exposure but may repress that response over time due to hormesis (as described above). Since their LINE1 results and the use of chronic exposure to heat does not exclude the possibility that transposons could be activated with heat treated spermatocytes, the authors should soften and hedge their conclusions in this section.

6.) Focus of paper shifts significantly from the intro to the end discussion. The manuscript starts on a markedly clinically footing and ends with a lot of conjecture about evolutionary function and development of temperature-dependent mammalian spermatogenesis. The paper would benefit by bringing the narrative full circle at the end and paring down the final paragraph.

Specific comments:

1. Line 86-87: The use of "non-invasively" when scoring progression of spermatogenesis within explants is unclear. Please clarify
2. Lines 158-159 – to make the paper more accessible to a non-mouse meiosis audience, the authors should make it clear here (when they're first brought up) what specific stages of meiotic prophase I SCP3 and SCP1 localization indicate.
3. Lines 162-163: The authors state that "essentially no pachytene or diplotene spermatocytes were observed". The authors should give more details about how this was determined (e.g. by eye or objectively assessed/quantified). Although this is in the Methods section, the % of nuclei surveyed demonstrated complete synapsis in each group should be indicated in the figure legend.
4. Lines 193-194: "...at which homology-dependent mismatch repair between homologous chromosomes establishes the synapsis and crossing over" needs clarification. A lot is covered in that sentence, please focus on the role the marker DMC1 plays/what meiotic process it is marking.
5. Line 234-236: "It is suggested therefore that a high temperature condition primarily impairs DSB repair, causing asynapsis, which in turn elicits apoptosis by meiotic checkpoint upon zygotene-pachytene transition." The authors should amend this statement to read as such: "It is suggested therefore that a high temperature condition impairs the loading of early recombination proteins or impairs DSB repair by homologous recombination, produces asynapsis, and elicits apoptosis..."
6. Lines 239-242: Sentence beginning with "Unexpectedly, we found..." requires editing to clarify the meaning of the statement.
7. Line 242: It is not a generally held view that the effects of heat on spermatogenesis are gradual and non-specific due to the effects of hormesis (see major comment above). There is a large body of literature dedicated to defining the specific mechanisms underlying heat-induced male infertility.
8. Lines 258-262: The description of the meiotic checkpoint mechanism primarily as a "guardian of species" seems misplaced. The meiotic checkpoint is generally accepted to play a role in preventing the formation of potentially defective gametes in wildtype contexts. This work demonstrates a role for this checkpoint in eliminating heat-damaged spermatocytes specifically, however that is not a novel role for this mechanism. For example, the CHK2-dependent checkpoint monitors both DNA damage and asynapsis such that extensive asynapsis leads to oocyte loss by inhibiting HR repair rather than triggering a distinct "synapsis checkpoint" (Rinaldi et al, Mol Cell 2017). The authors should amend their statements accordingly.
9. This suggestion is outside the scope of the paper but it's just a suggestion for future use and development of this system. Since spermatogenesis in this ex vivo system is slower than in vivo, the studies and conclusions on meiotic progression made with this system would be strengthened by adapted the system to synchronize spermatogenesis using retinoic acid as has been done in

studies by David Page's lab and Francesca Cole's lab.

Reviewer #3 (Remarks to the Author):

In this manuscript, the authors study the effect of heat on mouse spermatogenesis. To do so, they used an ex vivo culture system previously developed by the same team that allows spermatogenesis to occur in vitro. Using this system, the authors identify that, as it occurs in vivo, the temperature that allows proper spermatogenesis progression is 34°C. Also, they report that temperatures higher than 34°C alter spermatogenesis in a stepwise manner, suggesting the existence of multiple temperature-sensitive steps in spermatogenesis. Finally, they report how a temperature of 38°C alters meiotic recombination leading to the activation of apoptosis and the consequent loss of spermatocytes.

Overall, this is a well-written manuscript in which the study design is clearly explained and the conclusions raised in the study seem to be supported by the results. Nonetheless, I think the manuscript would significantly benefit from a more detailed explanation of the data obtained in the experiments performed. Below is a list of comments/suggestions that I would like the authors to address before resubmitting the study to be considered for publication.

- Consider changing the title for a more descriptive one.
- Please add in Ext. Data Figure 1 representative images of the testis at time=0 of culture (from 4 days old mice).
- Was the evaluation of the progression of the testis cultures performed blindly? If not, state why and mention it in the methods section.
- Please provide a quantification of the number of the different spermatogenic cell types found in the different culture conditions evaluated (Fig. 2).
- Since all experiments have been performed with more than one sample, please report the results as mean and standard deviation found in the different replicates (Fig. 2d and 3b).
- Fig. 4a: It seems that cultured samples may have residuals gH2AX foci at diplotema (34°C) and pachynema (37°C), which is a common sign of recombination defects in mutant mice. Thus, I would urge the authors to quantify the number of gH2AX foci at these stages in all samples analyzed.
- I have noticed that the DMC1 foci count in early leptotema for samples cultured at 37°C and 38°C are different from the one found at 34°C. While the sample cultured at 37°C has higher DMC1 counts, the one cultured at 38°C has lower counts. I would like the authors to discuss these data and why do they think the 37°C samples seem to go against what occurs at later stages.
- It is not clear to me how the authors think high temperature affects meiotic recombination.
- I think the DMC1 results are very interesting and I would also like to suggest the authors studied them a little bit more. The reduction of DMC1 numbers could be interpreted as spermatocytes produce fewer DSBs or alternatively they produce a normal number of breaks but these are repaired using DMC1-independent mechanisms (or DMC1 does not load). I think quantifying RAD51 and RPA (or MEIOB) foci in cultured samples would provide a much clearer image of how meiotic recombination is affected by raising the culture temperature.
- The fact that the proportion of pachytene spermatocytes is very low at 37°C makes me think that most spermatocytes may have significant amounts of unrepaired DSBs and these may be eliminated by surveillance mechanisms. Thus, I think it would be interesting to study the incidence of apoptosis in combination with DNA damage.
- Please provide a quantification of the asynapsis phenotype observed in samples cultured at 37°C.
- Page 12, lines 235-236. The authors seem to suggest that surveillance mechanisms only respond to asynapsis, while there are several studies that show that recombination errors are sufficient to activate an apoptotic response. Thus, I would suggest the authors reconsider the wording of this sentence.
- The data on the quails, although interesting, I think does not belong to this study and should be presented in a separate manuscript.
- Page 33, line 672. The number of cells counted stated here does not match with the ones in the figure.

Point-by-point responses to the comments by Reviewer #1:

In this study, the authors investigate the effects of temperature in germ cell differentiation and meiotic recombination in mouse testis explants (a system previously developed by the same lab). In this context, 34°C seemed to be the temperature most compatible with spermatogenesis and spermiogenesis, while lower (30°C, 32°C) and higher (35-40°C) temperatures blocked germ cell differentiation at different stages with both temperature extremities severely resulting in total germ cell depletion. The authors then focused in further exploring the underlying mechanisms for the observed meiotic prophase I blockage at 37°C and 38°C. They showed that at these temperatures, late spermatocytes (at pachytene and diplotene) are rare or absent. They then related this to an increase in the staining of apoptotic markers at 38°C comparatively to 34°C. Finally, the authors analyzed homolog synapsis and DMC1 foci numbers, as means to investigate meiotic recombination efficacy at 37°C and 38°C comparatively with *in vivo*- and *ex vivo*-grown (at 34°C) testes. The authors found synapsis defects in zygotene spermatocytes and a significant reduction in DMC1 foci numbers in late leptotene and zygotene spermatocytes at 37°C and 38°C. The authors then conclude that heat impairs meiotic recombination and synapsis of homologs which leads to a block in zygotene-pachytene transition and meiotic checkpoint-induced apoptosis.

Although the study is interesting, in my opinion the data shown is still quite preliminary for the claims that the authors want to make. In particular, the analyses of the impact of temperature in spermatocytes at the mechanistic level are superficial and need to be further explored. My specific comments are as follows:

We acknowledge that this study has drawn the Reviewer's potential interest. We are also grateful for his/her constructive comments to improve this study.

Comment 1-1. The culture of testis explants at 34°C seems to already lead to a significant decrease in cell density in the seminiferous tubules (Fig. 1d,e), suggesting that germ cells are already under some stress in these conditions. This could be further exacerbated by combining with the temperature shift in the explants. Thus, in my opinion the effects that the authors see with temperature shift should be taken with a grain of salt as a synergistic effect between *ex vivo* testis culture and temperature shift cannot be completely excluded based on the experiments of this study. To confidently support all the conclusions stated by the authors, the comparison with *in vivo* testes would be the most appropriate, although I understand that this would be unfeasible to do in a short time frame. Therefore, this aspect should be at least discussed in the manuscript.

We acknowledge this critical point. We agree that the current *ex vivo* culture supports spermatogenesis to an inferior extent compared with *in vivo*. However, the *ex vivo* culture provides an invaluable, otherwise unavailable setting to study the temperature dependence, because the testis temperature cannot be controlled accurately *in vivo*. In this study, taking such advantages of *ex vivo* study, we found the delicate temperature dependence of mouse spermatogenesis, with multiple steps impaired at different temperatures. Of note, this key finding is consistent with the *in vivo* observations that multiple cell types (e.g., spermatocytes and spermatids) are damaged and lost following surgical translocation of the testis to the abdomen. These advantages and disadvantages of the *ex vivo* study have been discussed in the revised manuscript (Lines 63-64, 310-312, 383-391).

Comment 1-2. It is unclear how the authors quantified the Acr-GFP signal in the testis explants (could not find any related information in the methods section). In this regard, the GFP signal in images of testis explants grown at 34°C (Extended Data Figure 1d) is not as uniformly distributed and no example fully recapitulates the signal of grade 4 in Extended Data Figure 1c, even though the majority of explants are grade 4 based on the quantifications (Figure 1b). Some regions of the 34°C-grown testis explants

have no GFP signal and it is unclear whether this is because there are no tubules. I would strongly suggest the authors to include in Extended Data Figure 1c images showing another marker for tubules and clarify how the number of tubules was taken into consideration when quantifying the Acr-GFP signal.

In the original manuscript, we have not detailed how spermatogenesis proceeds in our *ex vivo* culture but referred mainly to the previous papers (Sato et al., Nature 2011 and many others). In revision, we have provided the readers with information to evaluate the *ex vivo* observations in this paper.

Although our original statements might be lame regarding the grading method, our grading does not rely on counting the GFP+ and GFP- seminiferous tubules. Instead, we graded the explants by eye, based on the approximate fraction of Acr-GFP+ region out of the explant's periphery. In particular, explants were graded 0, 1, 2, 3, or 4, respectively, when GFP+ region covered 0% (i.e., no GFP signal detected), approximately 0-25%, 25-50%, 50-75%, or 75-100% of the periphery (Extended Data Fig. 1b). We applied such a semi-qualitative evaluation method primarily due to the limitation that grading must depend on instant and low-quality observation: Upon the weekly medium change, we could only observe the explants using an inverted fluorescent microscope, under optically poor conditions through the thick agar base. In addition, critically, the grading needs to be completed quickly to minimise the temperature fluctuation. However, despite its low resolution, this grading method provides consistent and reproducible results between explants cultured in independent experiments (many refs including Sato et al., 2011, Komeya et al., 2016, Sanjo et al., 2018).

The Reviewer also points that the example images in the original Extended Data Fig. 1c appear inconsistent with the grading shown for other experiments. In fact, the examples indicate the explants harbouring best-developed spermatogenesis within the range of each grade, a point that we should have described more explicitly in the original manuscript. Therefore, for example, the GFP signal may cover the entire periphery in some grade 4 explants but not in others. In revision, to avoid such confusion, we have rearranged this figure to correctly convey the grading criteria as mentioned earlier (new Extended Data Fig. 1b).

Regarding the third concern about the distribution of seminiferous tubules, the aforementioned grading method is justified because the entire periphery of an explant harbours well developed seminiferous tubules regardless of the culture temperatures (34°C, 37°C or 38°C), as observed in the newly added high-quality bright-field microscopic images (Fig. 1c and Extended Data Figure 1d). Broad distributions of the tubules were further verified in cross-sections after 5-week culture at different temperatures, stained with hematoxylin (Extended Data Figure 1e). These sections were also immuno-stained for Acr-GFP and MVH (a pan-germ cell marker) (Extended Data Fig. 1f). The Acr-GFP signals were consistent with the grading based on the afore-mentioned semi-quantitative methods. While, the broad distribution of MVH+ cells in all the peripheral seminiferous tubules, at varying densities, indicates germ cells' presence in all the peripherally-located seminiferous tubules, with varying numbers. These findings in the explant sections further support the validity of the grading methods. Related descriptions have been provided in the revised manuscript (Lines 86-94, 1013-1018).

Comment 1-3. Are the developmental stages of the seminiferous tubules shown in Fig1d,e, Fig2b,c, and Extended Data Figures 3 and 4 equivalent? This is important because different developmental stages of the seminiferous tubules have different proportions of distinct germ cell types which could affect the interpretation of these analyses.

Throughout this study, all the cultures (including those pointed by the Reviewer) started with testis explants prepared from 4-day-old mice. Although this point was described in the original manuscript (in the materials and method section), we realize it should have been emphasized more clearly. In revision, we have emphasized this critical information by presenting in a new figure panel (Fig. 1b) and enriching the related texts (Line 74-76, 84, 588-590).

Comment 1-4. I am not sure if I agree completely with the conclusions that the authors state in lines 137-141. The results in Figures 1 and 2 show that there is a positive correlation between early defects in germ cell differentiation and temperature increase. This can also be interpreted as a progressive aggravation of the same defects at early stages of germ cell development (for instance in homolog synapsis or DSB repair) such that a threshold is reached at higher temperatures resulting in a complete blockage of germ cell progression. In this respect many studies support a threshold in the number of recombination events needed to support successful chromosome pairing, synapsis, and segregation in mice (Kauppi et al., 2013; Faieta et al., 2016; Faisal et al., 2016). Furthermore, there is evidence supporting a threshold for unrepaired DSBs to activate the pachytene checkpoint (Pacheco et al., 2015). Supporting this view, the data in Figure 3b shows a progressive increase in the percentage of spermatocytes at early prophase and a concomitant decrease of late spermatocytes with the increase of temperature. Thus, I think these points should be taken into consideration by the authors and further discussed in the pa.

We appreciate this highly insightful comment. We realize that the original statements related to Fig. 1 and 2 may be misleading. In fact, this section never intended to describe the primary molecular defects occurring at different temperatures. Instead, the statements should have focused on the cellular outcome more clearly, i.e., which cells were preserved and which were lost (or reduced) at particular temperatures. In revision, we have rewritten the relevant section (Lines 117-141) carefully to clarify that these figures summarize the cellular defects but not the primary molecular defects.

Importantly, we agree with the Reviewer that the observed cellular defects may not reflect the primary impairment at the molecular level. Of note, the result of γ H2AX quantification performed in revision supports the insight given here by the Reviewer, significantly deepened the conclusions of this study: Briefly, at 37°C, we found that DSBs was increased at zygotene compared with 34°C due, at least in part, to the defective DSB repair (Fig. 4b-c and Fig. 5a-f). Some of these zygotene spermatocytes, harbouring DSBs and consequent asynapsis at levels above the threshold, are eliminated by the meiotic checkpoint. In contrast, others with DSBs and asynapsis below the threshold survived to the late pachytene stage. However, such surviving cells carried over higher levels of DSBs, leading to defective crossover formation (i.e., reduced MLH1 foci) at late pachytene and did not complete meiotic divisions. These observations and reasonings of the latent effect have been articulated in the text (Lines 210-217, 337-345) and a summary schematic (Fig. 7).

Comment 1-5. Related to the point 4, the authors should further comment on the fact that the decrease in the percentage of spermatocytes at pachytene and diplotene stages is accompanied by an increase in spermatocytes at early stages.

Indeed, Fig. 3b indicates the proportion of spermatocytes at each stage, but not the absolute number. Given the variable efficiency of *ex vivo* culture, we cannot conclude from these data if the increase in the percentages of leptotene and zygotene at 37°C and 38°C indicates an accumulation of these cells, or if it reflects the cell loss around the zygotene-pachytene transition.

Instead, as detailed for the following comment, we conducted quantitative analysis suggesting that, at 37 and 38°C, spermatocytes are “stuck” at an aberrant “asynapsed pachytene” state with characteristic unsynchronous homologue pairing, and eliminated by meiotic checkpoint (Fig. 6a-b). We hope that this would help to address the comment, albeit indirectly.

Comment 1-6. Given that the authors find synapsis defects in zygotene- and pachytene-like spermatocytes, I strongly recommend the authors to quantify the proportion of these abnormal cells found in each condition and differentiate these in the graphs of Figure 3b. Also, please clarify what criteria are used to stage the cells.

This is an excellent suggestion. In revision, we have categorized the aberrant spermatocytes frequently observed at high temperatures as “asynapsed pachytene”, using the criteria of harbouring one or more wholly synapsed chromosomes and one or more autosome pairs that are entirely asynapsed or are synapsed partly in less than 50% of its length, following previous studies (Ollinger et al., 2008 and Crichton et al., 2017).

It may be confusing verbally for non-specialists, but we clearly explained that “asynapsed pachytene” is classified as a subset of “zygotene” when we follow the generic definition used in Fig. 3a-b. Because, the generic criteria define “pachytene” when all autosome pairs have fully synapsed, and define “zygotene” when one or more autosomal pairs have partly synapsed, as described in Lines 250-266. The characterization and quantification of asynapsed pachytene spermatocytes are summarized in Fig. 6a-c.

We acknowledge that defining and quantifying *asynapsed pachytene* have significantly deepened our understanding of the spermatogenesis defects at high temperatures. In particular, as shown in Fig. 6d, the finding that *asynapsed pachytene* spermatocytes are devoid of MLH1 foci (a hallmark of late pachytene) provided vital insights as follow. All spermatocytes at 38°C, and a considerable fraction at 37°C, are stuck at an aberrant, *synaptic pachytene* state, eliminated by meiotic checkpoint without proceeding to late pachytene. These reasonings have been articulated in Lines 270-275, 333-341 and schemed in the new Fig. 7.

Comment 1-7. The authors only use one marker for transposons. The use of a second marker would strengthen the results. The authors should also comment on the increased levels of transposon activity of 34°C-grown explants compared to in vivo grown testis.

We agree that other transposons than LINE1 warrant investigation. However, given the main conclusion of this study that compromised DSB repair is a key heat-vulnerable process, analyzing other transposons would go beyond the scope of this study. Instead, as suggested by other Reviewers and the Editor, we have largely re-written the text to be much less conclusive. In particular, we emphasized it remains an open question for future study if other transposons contribute to the increase of DSBs at high temperatures like *C. elegans* (Kurhanewicz et al., 2020) (Line 372-378).

In addition, we have deepened the relationship between LINE1 upregulation and spermatogenesis defect. Using an *in vivo* cryptorchid assay, we have shown that LINE1 is not upregulated in surgically translocated testes in the abdomen, even just before spermatocytes start to die (Extended Data Fig. 5a-b). This observation is consistent with the conclusion that LINE1 expression is not causal for the increase of DSBs and subsequent cell death in spermatocytes, as described in Lines 295-305.

On the other hand, as pointed by the Reviewer, LINE1 is upregulated in explants even at 34°C, from unknown mechanisms. This phenomenon might be involved in the less effective spermatogenesis progression *ex vivo*, as discussed in Lines 388-391.

Comment 1-8. I recommend that more pachytene cells are analyzed for the 37°C condition in Figure 4c because they seem to have significantly less DMC1 foci as shown in Extended Data Figure 5 (here no DMC1 foci is seen which is quite different from the in vivo or 34°C conditions).

We have analysed more pachytene spermatocytes in Fig. 5f as suggested. The results reinforced that the number of DMC1 foci did not show a statistically significant difference. As for Fig 5e, we realise that the presented example was not so representative: we have replaced it with a more representative example.

comment 1-9. In lines 259-261, the authors state that “This mechanism, discovered and analysed intensively using genetic mutants, plays a role as a guardian of species in inhibiting gametogenesis in interspecific hybrids (a phenomenon known as hybrid sterility)”. I am not sure of what the authors mean by this. Extensive synapsis defects is thought to be the underlying cause of infertility in hybrid mice (Bhattacharyya et al., 2013). Moreover, several published studies have shown that the infertility seen in some mouse hybrid strains is caused by incompatibility between the Prdm9 allele of one strain and the genome of the other, such that PRDM9 asymmetric DNA binding affects the efficacy of homolog synapsis (Davies et al., 2016; Mihola et al., 2009; Flachs et al., 2012, Balcova et al., 2016). I recommend that the authors clarify this part and comment on their results in the light of these evidence.

We appreciate for pointing this issue, which may be confusing and misleading. In revision, the related part has been significantly weakened and shortened to focus only on the following reasoning (Lines 346-349). Here, we just state that hybrid sterility is a known role of the meiotic checkpoint; and that this study suggests that eliminating heat-damaged spermatocytes is one of the native roles of the meiotic checkpoint.

Comment 1-10-1. The conclusion that the authors make regarding defects in meiotic recombination repair by temperature shift merely based on the decrease in DMC1 foci numbers is faulty. A decrease in DMC1 foci could also result from a decrease in the number of DSBs (as γ H2AX signal was not quantified it is unclear whether this is the case), or even in a decrease in DMC1 protein levels. Thus, I strongly recommend the authors to quantify γ H2AX signal in spermatocytes in the different conditions, as well as the foci number of other DSB markers such as RPA or MEIOB, and RAD51 in these cells.

This is also an insightful and constructive comment. As suggested, we have conducted a series of quantifications. First, we quantified the γ H2AX signal intensity and found that DSBs were significantly increased at 37°C and, more prominently, 38°C (Fig. 4b-c). We also quantified the RPA (RPA2, in particular) foci, which were found not to change greatly at 37°C and 38°C (Fig. 5a-b). Further, we also found that the number of RAD51 foci reduced at 37°C and 38°C, consistent with the DMC1 foci shown in the original manuscript (Fig. 5c-d).

These new observations reinforce the conclusion that DSB repair machinery is heat-vulnerable. Further, they also provide insights into how DSB repair is compromised at high temperatures. First, the increased DSBs and roughly unchanged RPA foci imply that the positioning of RPA to DSBs are impaired at high temperatures. Second, the decrease of RAD51 and DMC1 foci (relative to the RPA foci) indicates that recruitment of RAD51 and DMC1 to RPA-marked DSBs are also impaired. These reasonings have been provided in Lines 221-241.

comment 1-10-2. If feasible, western blotting of DMC1 and/or RAD51 to analyse their protein levels should be done as high temperatures could destabilise these proteins.

Unfortunately, Western blotting is not feasible technically. The content of spermatocytes in explants is small and variable; collecting spermatocytes of particular stages (e.g., zygotene) for Western is practically impossible. Instead, we have discussed that the reduction of the RAD51, DMC1, and RPA2 foci could involve a reduced amount of these or other proteins (Lines 361-362).

Comment 1-10-3. It would also be interesting to analyze the foci of downstream repair factors, such as MSH4 or MLH1, to further understand the impact of temperature in the resolution of non-crossovers/crossovers, and what proportion of the few 37°C-grown pachytene spermatocytes (Figure 3b) efficiently repair DSBs and form normal numbers of crossovers.

We appreciate this insightful suggestion, too. In revision, we have quantified the MLH1 foci in spermatocytes that had developed *ex vivo* at different temperatures, shown in Fig. 6d-g. The results and relevant insights are summarized below, which have been articulated in Lines 266-281, 337-345 and schemed in the new Fig. 7.

In *in vivo*-grown testes, as expected, the numbers of MLH1 foci per pachytene spermatocyte showed a bimodal distribution: Some are devoid of MLH1 foci while many others harboured about 22-30 foci, representing early and late pachytene, respectively. A few contained intermediate numbers of foci, likely during the foci-forming process proceeding quickly and synchronously.

At 34°C, fully synapsed pachytene showed a similar bimodal distribution, with the foci-containing group showing a slight reduction to about 20-27 foci per nuclei. These findings are consistent with the apparently normal progression of meiotic prophase I.

At 38°C, as briefly discussed earlier (comment 1-6), spermatocytes fell into the *asynapsed pachytene* state and underwent cell death. Strikingly, no MLH1 foci were observed in *asynapsed pachytene* even on the fully synapsed chromosomes. Given that MLH1 foci normally form in late pachytene spermatocytes (after stage IV), which have survived beyond the meiotic checkpoint, this observation suggests that *asynapsed pachytene* is eliminated at the meiotic checkpoint. This conclusion is further supported by the absence of *Acr-GFP* expression, which turns on at stage IV pachytene, at 38°C.

At 37°C, some severely affected spermatocytes fell into *asynapsed pachytene*, which developed no MLH1 foci and died at the meiotic checkpoint, like those observed at 38°C. In contrast, other, less affected cells survived to late pachytene to bear MLH1 foci. However, the number of MLH1 foci was further reduced to about 17-25 (Figure 6 d-f). Given the 19 autosome pairs and one pair of sex chromosomes, pachytene spermatocytes at 37°C often included fully synapsed chromosomes without MLH1 foci (Fig. 6d). Thus, crossover formation is critically compromised in these cells, failing to complete meiotic division.

Comment 1-11. Related to point 10, it seems that increased levels of γ H2AX are present at late spermatocytes grown at 37°C and 38°C. Thus, providing the γ H2AX quantification in these cells will also clarify this which may further support defects in DSB repair.

We have quantified the γ H2AX signal (Fig. 4b-c), and verified higher levels of DSBs in spermatocytes grown at high temperatures. We refer to **Comment 1-10-1**, to avoid repetition.

Comment 1-12. In the sentence “At 36°C, although similarly intense *Acr-GFP* signal was observed, elongating spermatids were absent, and a few round spermatids were observed as the most advanced cell type” (lines 125-127) it is not clear to what condition is it comparing to.

This statement was based on the comparison with those at 34°C, and we have clarified this point (Lines 128-130).

Comment 1-13. The use of the terms “synapse formation” (line 151) or “synapse disassembly” (line 153) do not sound right. I would recommend to change it to, for example, “establishing synapsis” and “synaptonemal complex disassembly”.

Following this helpful comment, we have rephrased these terms (Lines 153-154, 158-159, 243-244).

Comment 1-14. “Leptotene spermatocytes...” (line 208) should be late leptotene spermatocytes.

As suggested, this has been rephrased (Lines 231-232).

Comment 1-15. The term “homology-dependent mismatch repair” (lines 193) is confusing. Please rephrase it.

As suggested, we have rephrased this term to “homology-dependent DNA repair” (Lines 192-193).

Comment 1-16. The way the authors describe the activation of the meiotic checkpoint, best known as the pachytene checkpoint, sounds as they are providing novel findings. It is already well established in the field that there is a mid-pachytene checkpoint that is activated in response to both asynapsis and DSB repair defects. I would recommend to rephrase this section and refer to appropriate studies in literature.

We appreciate this critical comment. We had no intention to say that a meiotic checkpoint is a novel finding. Instead, we would like to state that this well-known surveillance mechanism eliminates heat-damaged spermatocytes. We have rewritten this part to avoid misleading (Lines 318-322, 329-330, 346-349).

Comment 1-17. The authors often refer to the elimination of spermatocytes on zygotene-pachytene transition, however this is not completely correct. Although defective spermatocytes have incomplete synapsis, they are in a pachytene-like state and they are eliminated by the pachytene checkpoint, as is the case of other mutant spermatocytes (e.g., Pittman et al., 1998; Yoshida et al., 1998; Hamer et al., 2008).

We appreciate this critical comment, too. In response, we defined what the Reviewer refers to as a “pachytene-like state” as “*asynapsed pachytene*” using cytological criteria, as detailed in **comment 1-6**. Defining the *asynapsed pachytene* state substantially improved the resolution of analysing the process of meiotic failure, supporting the scenario articulated by the Reviewer. We refer to **comment 1-6** for the details and relevant figures and descriptions to avoid repetition.

Point-by-point responses to the comments by Reviewer #2:

Spermatogenesis is very temperature sensitive and heat exposure can result in male infertility. How temperature specifically affects the development of sperm is an emerging field. This manuscript used an *ex vivo* mouse testes explant preparation to assess the deleterious effects of chronic heat exposure on spermatogenesis, allowing for the parsing out of heat-sensitive mechanisms specific to the spermatogenic/meiotic program in the absence of potentially confounding physiological inputs. Additionally, the use of chromosome spreads allowed for a more thorough evaluation of the effects of heat on specific meiotic stages. The authors present data that support their claim that relatively small temperature elevations can impair specific stages of spermatogenesis, likely via stage-specific mechanisms. The authors clearly outlined the clinical relevance of this work and compared their specific results back to clinical histological findings to better dissect the mechanisms underlying specific health conditions (e.g. cryptorchidism vs varicoceles).

We acknowledge that the Reviewer has captured the central aim of this study, the advantage of the use of *in vitro* explant culture, and the potential significance of the outcome. We believe that revisions we made in response to his/her comments have significantly improved this study.

Comment 2-1. The synaptonemal complex is well documented in multiple systems to be temperature sensitive (reviewed in Morgan CH et al., *Phil Trans R Soc B* 2017; Bilgir et al., *G3* 2013; Loidl *Chromosoma* 1989). Although the SC components SCP1 and SCP3 are traditionally used markers to differentiate the meiotic prophase I stages of unsynchronized spermatocytes, SCP1/3 are also likely to be temperature sensitive, making them unreliable markers for staging of spermatocytes. Using only SCP1/3 to stage spermatocytes weakens the authors' conclusion that chronic heat treatment of spermatocytes causes arrest by zygotene. The authors should use alternative markers for staging their spermatocytes such as MLH1, which loads in pachytene, or perform FISH to assess pairing.

As pointed by the Reviewer, given that synaptonemal complex components can alter their structure and localisation in a temperature-sensitive manner in other organisms (e.g., *C. elegans* and *Arabidopsis*), the localisation of the mouse SCP3 and SCP1, which we rely on in evaluating the chromosome pairing and categorising the stages of meiotic prophase I, warrants verification at high temperatures.

In this regard, first, we emphasise that the patterns of SCP1 and SCP3 staining at 37°C and 38°C looked consistent with those in *in vivo* and 34°C samples, as described in the original manuscript. In particular, SCP3 (an axis component) was localised in both paired and unpaired parts, while SCP1 (an SC component) paints the paired parts only. These findings support the reliability of SCP1 and SCP3 staining.

We further evaluated the reliability of our staging using two additional markers. First, we compared the localisation SCP1 and SCP3 with HORMAD1, another axial element protein localised to unpaired parts but not to the synaptonemal complex. In spermatocytes grown at 37°C and 38°C, we found that HORMAD1 and SCP1 showed consistent, reciprocal localisations in unpaired and paired parts, respectively, consistent with *in vivo* and 34°C samples. (Extended Data Fig. 3a-b, and descriptions in Lines 160-165). Second, as explicitly suggested by the Reviewer, we also conducted MLH1 staining to evaluate the crossovers (Fig. 6d). Again, showing consistent localisation on fully synapsed pachytene chromosomes, MLH1 staining has provided in-depth information about the crossover formation at different temperatures. We refer to **Comment 1-10-3** for details to avoid repetition.

In addition, we discuss the temperature sensitivity of the synaptonemal complex reported in other organisms as a possible mechanism of heat vulnerability of mouse male meiosis (Lines 363-364).

Comment 2-2. Multiple studies have demonstrated DMC1 only loads to a subset of DSBs in mouse spermatocytes and that RAD51 will load to DSBs not destined for homologous recombination in spermatocytes. Based on unquantified H2AX staining and quantified DMC1 staining, the authors conclude that DSB repair is impaired (line 235). This data implies that homologous recombination may be affected (and not that DSB repair in general is impaired). To make this conclusion, the authors need to do the following: 1) quantify H2AX staining; 2) assess/quantify RAD51 staining; and, 3) assess/quantify MLH1 staining (which will determine if homologous recombination is indeed impaired – because MLH1 marks the downstream crossover site that forms in pachytene).

In response, we have conducted the three suggested analyses. First, as described in **comment 1-10-1**, through quantifying the γ H2AX signal in late leptotene, we found significantly higher levels of DSBs at 37°C and 38°C, compared to *in vivo* and 34°C (Fig. 4b-c). Second, we found that RAD51 foci were reduced at 37°C and 38°C compared with *in vivo* and 34°C in late leptotene, in parallel with DMC1 foci (Fig. 5c-f).

Further, by MLH1 staining, we found that “asynapsed pachytene” spermatocytes observed at 37°C and 38°C were devoid of MLH1 foci, suggesting that crossover formation does not proceed in this aberrant state (Fig. 6d-g). As detailed in **comment 1-10-3**, we also found that the fully synapsed pachytene spermatocytes at 37°C harboured a reduced number of MLH1 foci (ranging from about 17-25, relative to the 20 chromosome pairs), some harbouring chromosome pair(s) without MLH1 foci indicating compromised crossover formation (Fig. 6d-g). No spermatocytes survive to the late pachytene stage at 38°C.

Based on these new findings, we conclude that DSB repair in the early and late stages, with and without homologous recombination between homologous chromosomes to form crossover, respectively, are compromised at high temperatures. These reasonings have been described in Lines 207-220, 266-281 and summarized schematically in Fig. 7.

Comment 2-3. Figure 4C uses a Mann-Whitney U test (non-parametric) to compare 2 sample sets. Performing several Mann-Whitney tests, without a correction, can give spurious results. To compare 2 or more groups, it is more appropriate to run Kruskal-Wallis ANOVA which has the same assumptions as M-W test.

Thanks to this helpful suggestion, we have performed Kruskal-Wallis ANOVA to data in Fig. 5f (the number of DMC1 foci) and newly added counts for RPA2, RAD51 and MLH1 foci (Fig. 5b, Fig. 5d and Fig. 6g).

Comment 2-4. There are differences in the short-term and long-term responses to either chronic or acute heat stress. Specifically, chronic heat stress may elicit a hormesis effect (or biphasic response), therefore an effect may only occur during the initial exposure but then the system adjusts with time (see next comment regarding discussion of transposons). The manuscript addresses the effects on chronic heat stress on spermatogenesis after several days/weeks of heat treatment. The authors should clearly indicate and discuss their results/conclusion in the context of chronic heat stress effects and the effects they may be missing due to hormesis.

This is a really important and critical comment. We agree with the Reviewer in that, during the chronic temperature elevation, the initial (short-term) and late (long-term) effects may be different, or even opposite, due to the “hormesis effect”. This point has been emphasized in the revised manuscript (Lines 300-303, 379-382). Regarding the specific issue about transposon expression, please see the next **comment 2-5**.

Comment 2-5. Lines 179-187: Using LINE1 alone, the authors conclude that transposons are not activated upon chronic exposure to heat. Several models have recently demonstrated that heat exposure can activate specific classes (or subsets) of transposons and not others (Gusa et al., PNAS 2020; Kurhanewicz et al., Current Biology 2020). Further, transposons could be activated at the onset of heat exposure but may repress that response over time due to hormesis (as described above). Since their LINE1 results and the use of chronic exposure to heat does not exclude the possibility that transposons could be activated with heat treated spermatocytes, the authors should soften and hedge their conclusions in this section.

We acknowledge this insightful comment related to the previous one. We fully agree that the potential involvement of transposons must have been discussed deeper. Notably, a recent study in *C. elegans* has demonstrated that acute heat shock (34°C for 2 hours, compared to the standard temperature condition of 20°C) induces transposon expression and an increase of RAD51 foci in spermatocytes.

Motivated by this and previous comments and these prior studies, we revisited the analysis of LINE1 transposon expression in our chronic *ex vivo* experiments. In particular, while the temperature-dependence of LINE1 expression had not been observed in the long term (after 5 weeks of culture), we questioned if LINE1 might be activated in the short term after the temperature shift. To this end, we used an artificial cryptorchid experiment *in vivo*: We asked if LINE1 is upregulated 2 days after testis translocation, a time point when spermatocytes must be damaged already since they started to die on the next day. We observed no upregulation of LINE1 in spermatocytes and other germ cells (Extended Date Fig. 6a-b).

This result reinforces our original notion that spermatogenesis defect following temperature elevation does not involve dysregulation of LINE1. However, we do not, and cannot, exclude the involvement of other transposons at all. These reasonings and reservations have been described in the revised manuscript (Lines 295-305, 372-378).

Comment 2-6. Focus of paper shifts significantly from the intro to the end discussion. The manuscript starts on a markedly clinically footing and ends with a lot of conjecture about evolutionary function and development of temperature-dependent mammalian spermatogenesis. The paper would benefit by bringing the narrative full circle at the end and paring down the final paragraph.

We appreciate this constructive suggestion. We have extensively rearranged this manuscript to achieve a linear description. We also removed all descriptions about the avian (quail) spermatogenesis, as suggested by another Reviewer (**comment 3-13**). We hope that the structure of the paper has improved considerably.

Comment 2-7. Line 86-87: The use of “non-invasively” when scoring progression of spermatogenesis within explants is unclear. Please clarify

We meant, by this phrase, that the same explants can be graded under a microscope over the culture period, without fixation or other processing. This part has been rephrased in revision for clarification (Lines 58-59).

Comment 2-8. Lines 158-159 – to make the paper more accessible to a non-mouse meiosis audience, the authors should make it clear here (when they’re first brought up) what specific stages of meiotic prophase I SCP3 and SCP1 localization indicate.

In response to this constructive suggestion, we tried to describe as clearly as possible the localisation of the mouse SCP3 and SCP1 in each stage of meiotic prophase I in Lines 145-165.

Comment 2-9. Lines 162-163: The authors state that “essentially no pachytene or diplotene spermatocytes were observed”. The authors should give more details about how this was determined (e.g. by eye or objectively assessed/quantified). Although this is in the Methods section, the % of nuclei surveyed demonstrated complete synapsis in each group should be indicated in the figure legend.

Following this comment, we have clarified that this classification is based on visual observation in the legend of Fig. 3 (Lines 635-637). We also indicated the percentages of spermatocytes that are completely synapsis (those classified to pachytene and diplotene) (Lines 641-644).

Comment 2-10. Lines 193-194: “...at which homology-dependent mismatch repair between homologous chromosomes establishes the synapsis and crossing over” needs clarification. A lot is covered in that sentence, please focus on the role the marker DMC1 plays/what meiotic process it is marking.

In response to this helpful suggestion, we have broken down this sentence, focusing on the role of DMC1, along with those of RAD51 and RPA, in leptotene and zygotene (Lines 190-193).

Comment 2-11. Line 234-236: “It is suggested therefore that a high temperature condition primarily impairs DSB repair, causing asynapsis, which in turn elicits apoptosis by meiotic checkpoint upon zygotene-pachytene transition.” The authors should amend this statement to read as such: “It is suggested therefore that a high temperature condition impairs the loading of early recombination proteins or impairs DSB repair by homologous recombination, produces asynapsis, and elicits apoptosis...”

As we have made substantial rewriting and rearrangements of the manuscript, the contents of this summary sentence were moved from the bottom of the result section to the discussion, expanded and reorganised into multiple sentences across paragraphs (Lines 329-345, 355-364). Although the Reviewer’s suggestion has not been reflected in the revised version verbally, we believe that we have articulated what the Reviewer would like to include. If we have not, we would be happy to do further rewriting.

Comment 2-12. Lines 239-242: Sentence beginning with “Unexpectedly, we found...” requires editing to clarify the meaning of the statement.

In response to this and the next comments, we have rephrased this sentence concisely and neutrally as “We found that spermatogenesis is impaired at multiple steps in a strict temperature-dependent manner” (Lines 307-309).

Comment 2-13. Line 242: It is not a generally held view that the effects of heat on spermatogenesis are gradual and non-specific due to the effects of hormesis (see major comment above). There is a large body of literature dedicated to defining the specific mechanisms underlying heat-induced male infertility.

We agree that our original statement was misleading, given the hormesis effect and other mechanisms described for the heat damage of spermatogenesis. We have deleted this sentence in the revised manuscript.

Comment 2-14. Lines 258-262: The description of the meiotic checkpoint mechanism primarily as a “guardian of species” seems misplaced. The meiotic checkpoint is generally accepted to play a role in preventing the formation of potentially defective gametes in wildtype contexts. This work demonstrates a role for this checkpoint in eliminating heat-damaged spermatocytes specifically, however that is not a novel role for this mechanism. For example, the CHK2-dependent checkpoint monitors both DNA damage and asynapsis such that extensive asynapsis leads to oocyte loss by inhibiting HR repair rather than triggering a distinct “synapsis checkpoint” (Rinaldi et al, Mol Cell 2017). The authors should amend their statements accordingly.

We are sorry that our original statements have caused such confusion. We had no intention to claim that we discovered that meiotic checkpoint plays a role in the wild-type context. Instead, as the Reviewer points, it is established that the meiotic checkpoint is activated by asynapsed chromosomes and unrepaired DSBs, both in wild type and mutants, irrespective of the initial causal factors. This point has been emphasized explicitly as “Meiotic checkpoint surveys primary spermatocytes with abundant DSBs and asynapsis, prevents the damaged cells from progressing to late pachytene, and eliminates them through apoptosis. Here, what causes DSBs and asynapsis does not matter” (Lines 320-332).

Further, we have articulated (Lines 329-345) and schemed (Fig. 7) the implied scenario of eliminating the damaged cells in which the meiotic checkpoint plays vital roles, without sentences that might sound like we had discovered that the meiotic checkpoint eliminates the heat-damaged spermatocytes.

Comment 2-15. This suggestion is outside the scope of the paper but it’s just a suggestion for future use and development of this system. Since spermatogenesis in this ex vivo system is slower than in vivo, the studies and conclusions on meiotic progression made with this system would be strengthened by adapted the system to synchronize spermatogenesis using retinoic acid as has been done in studies by David Page’s lab and Francesca Cole’s lab.

In our *ex vivo* culture, the speed of spermatogenesis progression is not slower but comparable with the *in vivo* time course, as we reported in a previous study (Sato et al., Nature 2011). We have clarified this point explicitly in the revised manuscript (Lines 76-77). In any case, however, since we analysed the spermatogenesis after long enough culture (i.e., 5 weeks), slight delay or acceleration, if there, should not have affected the results of this study.

We appreciate the suggestion to synchronise the spermatogenesis progression in explants by manipulating the RA signal. We would consider this suggestion for one of the future experiments.

Point-by-point responses to the comments by Reviewer #3:

In this manuscript, the authors study the effect of heat on mouse spermatogenesis. To do so, they used an ex vivo culture system previously developed by the same team that allows spermatogenesis to occur in vitro. Using this system, the authors identify that, as it occurs in vivo, the temperature that allows proper spermatogenesis progression is 34°C. Also, they report that temperatures higher than 34°C alter spermatogenesis in a stepwise manner, suggesting the existence of multiple temperature-sensitive steps in spermatogenesis. Finally, they report how a temperature of 38°C alters meiotic recombination leading to the activation of apoptosis and the consequent loss of spermatocytes.

Overall, this is a well-written manuscript in which the study design is clearly explained and the conclusions raised in the study seem to be supported by the results. Nonetheless, I think the manuscript would significantly benefit from a more detailed explanation of the data obtained in the experiments performed. Below is a list of comments/suggestions that I would like the authors to address before resubmitting the study to be considered for publication.

We appreciate the Reviewer's potentially high evaluation of this study and his/her constructive suggestions, which we have addressed as detailed below.

Comment 3-1. Consider changing the title for a more descriptive one.

In response, we have changed the title to stress a more mechanistic aspect of heat sensitivity of mouse spermatogenesis as *"Temperature sensitivity of DNA double-strand break repair underpins heat-induced meiotic failure in mouse spermatogenesis."*

Comment 3-2. Please add in Ext. Data Figure 1 representative images of the testis at time=0 of culture (from 4 days old mice).

As suggested, we have provided the images as suggested (Fig. 1b).

Comment 3-3. Was the evaluation of the progression of the testis cultures performed blindly? If not, state why and mention it in the methods section.

We have not conducted a blind evaluation of the spermatogenesis progression in ex vivo culture. This evaluation is performed visually by microscopic observation of the Acr-GFP fluorescence, when explants are taken out of the incubators set strictly at certain temperatures, at the timing of the weekly medium change. Therefore, observation must be completed in a short period to minimize the temperature fluctuation (cf. **Comment 1-2**). Further, critically, it is practically impossible for us to keep the culture temperature blind. Since, to do so, we must evaluate without knowing which incubator the culture dish under observation came from, without confusion in the routine procedure of medium change. This reasoning has been described in the Methods section (Lines 779-782).

Comment 3-4. Please provide a quantification of the number of the different spermatogenic cell types found in the different culture conditions evaluated (Fig. 2).

Although this is a sensible comment, we cannot be confident in presenting reliable cell numbers in each differentiation step based on the explant sections. Indeed, the density of spermatogenic cells varies considerably among the seminiferous tubules within an explant and between different explants.

Nevertheless, we found that the most advanced cell type observed at each temperature was strikingly reproducible between explants and experimental batches. This is why we still believe that documenting the presence or absence of individual cell types, along with their rough abundances in a qualitative manner, is the best method to describe the spermatogenesis progression in Fig. 2d and Extended Data Table 1.

Comment 3-5. Since all experiments have been performed with more than one sample, please report the results as mean and standard deviation found in the different replicates (Fig. 2d and 3b).

As for Fig. 2d, as described above (**comment 3-4**), we cannot show quantified frequencies of cells at each step in a reliable manner. However, this summary is based on observing multiple (many, indeed) samples for each temperature. The numbers of explants analyzed and those showing the representative features are shown in Extended Data Table 1.

In the case of Fig. 3b, as many as 6-7 explants were pooled for each temperature condition. Otherwise, we could not obtain enough number of well-prepared nuclear spreads without deformation, fragmentation, or merger with other nuclei, at each category (e.g., leptotene, zygotene, pachytene and diplotene). Therefore, we could not assess the explant-to-explant variation, and the values indicate the effective averages of several samples. In revision, these explanations have been provided in the legend (Lines 637-639) and Methods section of supplementary Information (Lines 826-828).

Comment 3-6. Fig. 4a: It seems that cultured samples may have residuals γ H2AX foci at diplonema (34°C) and pachynema (37°C), which is a common sign of recombination defects in mutant mice. Thus, I would urge the authors to quantify the number of γ H2AX foci at these stages in all samples analyzed.

In response to this suggestion, we first presented single-channel images of γ H2AX signal distribution in the revised manuscript for a better presentation of the γ H2AX distribution (Fig. 4b). Here, as the Reviewer points, traces of γ H2AX signals are often observed on synapsed chromosomes (e.g., pachynema) or as dots outside the chromosomal axes. However, these γ H2AX signals are irregular in shape and size, whose dull boundaries make it difficult to count the foci unambiguously. This is contrary to the regularity and discreteness of RPA2, RAD51, DMC1, and MLH1 foci.

Instead, to quantify the γ H2AX level appropriately, we have quantified the γ H2AX signal intensity from the acquired IF images (Fig. 4c). As detailed in **comments 1-10-1 and 2-2**, such quantifications have deepened our understanding of DSB levels at different temperatures (Lines 221-241).

Comment 3-7. I have noticed that the DMC1 foci count in early leptotene for samples cultured at 37°C and 38°C are different from the one found at 34°C. While the sample cultured at 37°C has higher DMC1 counts, the one cultured at 38°C has lower counts. I would like the authors to discuss these data and why do they think the 37°C samples seem to go against what occurs at later stages.

We appreciate such careful comments. In revision, we have re-evaluated the statistical significances using appropriate methods as suggested by Reviewer 2 (**comment 2-3**). As a result, among the numbers of DMC1 foci in early leptotene at different temperatures, we only found a weak significance between 34°C and 37°C ($p=0.04$), but not in all other combinations. Thus, to avoid overstatement, we would refrain from making conclusive discussions about its biological meaning.

Comment 3-8. It is not clear to me how the authors think high temperature affects meiotic recombination.

We have realized that this crucial issue has not been articulated explicitly in the original manuscript. Although the underlying molecular mechanisms remain an open question, we may hypothesize some molecular mechanisms: Overall, at high temperatures (e.g., 37°C or 38°C), the increased DSBs, and consequent asynapsis, together activate the meiotic checkpoint to induce apoptosis. While, the surviving cells at 37°C, with DSBs and asynapsis below the checkpoint threshold, still have a considerable amount of DSBs, leading to compromised crossover formation. Therefore, the increase of DSBs at zygotene is a key primary event in spermatogenesis failure at high temperatures.

Here, defects in the DSB repair must be crucial, given the decreased numbers of RAD51 and DMC1 foci. In particular, high-temperature conditions may impair the formation and/or maintenance of RPAs/RAD51/DMC1 foci at DSBs. We further suppose that interaction(s) between these proteins and the cut DNA may be temperature-sensitive. This hypothesis agrees with the decrease of RPA2 foci relative to γ H2AX signal and RAD51/DMC1 foci relative to RPA2 foci (Fig. 4b-c and 5). Alternatively, a larger-scale structural basis involving the aligned bivalent chromosomes and synaptonemal complex may change their ability to nourish the RPA2/RAD51/DMC1 foci, in a temperature-sensitive manner.

However, all these players are essential for both spermatogenesis and oogenesis. Therefore, these scenarios alone cannot explain the male-specific temperature sensitivity. Male-specific mechanism(s), e.g., additional sex-dependent protein components or chromosome structures, must be involved in the temperature sensitivity. The sex-dependence excludes the possibility that the stability of RPAs, RAD51 and/or DMC1 proteins *per se* is temperature-sensitive. Besides, the generation of DSBs could also be increased at high temperatures. Although it remains an open question if, and how, it happens, one possibility is the de-regulation of transposons, as recently demonstrated in *C. elegans*. Although our results do not support that LINE1 expression is dysregulated in response to high temperatures (Extended Data Fig. 5; **comment 2-5**), it remains an open question if other transposons are involved. These reasonings have been described in Lines 355-371.

Comment 3-9. I think the DMC1 results are very interesting and I would also like to suggest the authors studied them a little bit more. The reduction of DMC1 numbers could be interpreted as spermatocytes produce fewer DSBs or alternatively they produce a normal number of breaks but these are repaired using DMC1-independent mechanisms (or DMC1 does not load). I think quantifying RAD51 and RPA (or MEIOB) foci in cultured samples would provide a much clearer image of how meiotic recombination is affected by raising the culture temperature.

We acknowledge the interest shown by the reviewer. In revision, as suggested, we quantified the γ H2AX (DSBs) amount and RPA and RAD51 foci, in addition to the DMC1 foci at different temperature conditions. Although we would not repeat detailing here, the results suggest the elevation of DSBs, defective assembly of RPA and DMC1/RAD51, significantly deepened our insights about the meiosis failure at high temperature (Fig. 5 and description in Lines 221-241; cf **comments 1-4, 1-10-1, 2-2, 3-6 et al.**).

comment 3-10. The fact that the proportion of pachytene spermatocytes is very low at 37°C makes me think that most spermatocytes may have significant amounts of unrepaired DSBs and these may be eliminated by surveillance mechanisms. Thus, I think it would be interesting to study the incidence of apoptosis in combination with DNA damage.

This is an interesting suggestion. We also suppose that cells with a larger amount of DSBs may fall into an aberrant state of “asynapsed pachytene” and undergo checkpoint-mediated cell death, while those with less DSBs would survive to pachytene as discussed in the manuscript. However, it is technically difficult to address this hypothesis by experiments: While identification of asynapsed and fully synapsed pachytene spermatocytes and quantification of their DSBs require nuclear spread preparation, soluble

components are lost during this procedure, hindering reliable detection of apoptosis markers such as Caspase 3. Instead, we have articulated the aforementioned probable reasonings in Lines 329-345 and Fig. 7.

comment 3-11. Please provide a quantification of the asynapsis phenotype observed in samples cultured at 37°C.

This is also a vital and constructive suggestion. As detailed in **comment 1-6**, we have quantified “asynapsed pachytene” spermatocytes defined according to a previous study (Page et al., 2004 and Crichton et al., 2017). Namely, spermatocytes harbouring at least one completely synapsed homologue pair and at least one incompletely synapsed autosomes pairs exhibiting asynapsed regions over >50% of their length. The results are summarized in Fig. 6a-c and Lines 250-261.

comment 3-12. Page 12, lines 235-236. The authors seem to suggest that surveillance mechanisms only respond to asynapsis, while there are several studies that show that recombination errors are sufficient to activate an apoptotic response. Thus, I would suggest the authors reconsider the wording of this sentence.

In response to this very important comment, we have clarified that meiotic checkpoint is activated by both asynapsis and unrepaired DSBs (Lines 320-332).

comment 3-13. The data on the quails, although interesting, I think does not belong to this study and should be presented in a separate manuscript.

Following this suggestion, we removed the entire part and materials about the quail study. We believe that this removal, together with the addition of many new data in mice, has made this paper more focused.

comment 3-14. Page 33, line 672. The number of cells counted stated here does not match with the ones in the figure.

We have fixed this confusion. We appreciate pointing out this problem.

Reviewers' comments:

Reviewer #1 (Remarks to the Author):

The authors' efforts to answer all my concerns are much appreciated and the revised version of the manuscript is much improved. However, there are still some aspects that should be considered before publication. These are as follows:

- As the authors pointed out, it is quite confusing to include the "asynapsed pachytene" cells in the zygotene cell population. An alternative approach could be to include these in the pachytene cell population, or, if the authors prefer, consider them an independent population and call them abnormal/aberrant pachytene cells.

- The authors mention several times that "...RPA recognizes DSBs...". However, the most accurate way to say is "...RPA recognizes/binds to single stranded DNA...".

- Although in image 4b and text (lines 210-215) it claims higher γ H2AX in diplotene spermatocytes grown at 37C, the quantification does not represent that. Please rectify.

- In lines 226-227: "...the numbers of RPA2 foci were comparable to in vivo samples at all the temperatures tested." – This does not apply to zygotene stage where the RPA foci number decreases significantly at 34 and 37C relative to in vivo. Please rectify.

- Lines 228-229: "...we conclude that the DSB recognition by RPA was compromised at these temperatures." – An alternative explanation is that DSB resection is affected at higher temperatures. Please discuss this in the paper.

- Lines 275-278 are contradicting: "...a fraction of the (fully synapsed) pachytene spermatocytes surviving at 34C and 37C harboured MLH1 foci in numbers roughly comparable with those in vivo. However, their number reduced significantly to about 20-27 at 34C, and 17-25 at 37C...". Please clarify.

- Lines 278-279: "Given the 19 autosome pairs and a pair of sex chromosomes, some 37C grown pachytene spermatocytes include fully synapsed chromosomes lacking MLH1 foci (Fig. 6d-e). Thus, crossover formation is compromised in late pachytene developed ex vivo, subcritically at 34C, and critically at 37C." – perhaps, an alternative interpretation is that these pachytene spermatocytes that do not have MLH1 foci are spermatocytes that did not progress enough into pachytene given that COs (a.k.a. MLH1 foci) are known to be established in mid-to-late pachytene spermatocytes. Please discuss this in the paper. In this regard, and for future reference, many studies stain spermatocytes for the testis-specific histone H1 (H1t) to differentiate early from mid-to-late pachytene spermatocytes (e.g. Papanikos et al., 2019).

- Lines 55-57: the sentence "In *C. elegans*, transient heat shock causes dysregulation of transposon expression and DNA damage in spermatocytes." seems to be misplaced given that the next sentence (line 58) refers to testicular temperature.

- The English throughout the manuscript needs to be improved as in some instances it reads a bit slang or lacks clarity (some examples are listed below).

Line 7 : ... meiotic prophase I is damaged, which offers increased DNA...

Lines 8-10: Such damaged spermatocytes cause asynapsis between homologous chromosomes, eliminated through apoptosis at the meiotic checkpoint."

Line 215: ... spermatocytes showed higher γ H2AX levels, too."

Line 322: Here, what causes DSBs and asynapsis does not matter...

Lines 330-331: At 34°C, prophase I progresses largely normally despite...

Line 360: ...these proteins and the cut DNA may be compromised...

The abstract also needs to be significantly improved. I suggest the authors to resort to a professional editing service.

- Spread images are beautiful but in Figs.4b and 5 they are too small to be appreciated. Please

consider expanding these into new figures.

Reviewer #3 (Remarks to the Author):

The authors have properly addressed most of my comments, and the resubmitted version of the manuscript is significantly improved compared to the previous one. Thus, I have no objection to the publication of this study in Communications Biology.

Reviewer #1 (Remarks to the Author):

The authors' efforts to answer all my concerns are much appreciated and the revised version of the manuscript is much improved. However, there are still some aspects that should be considered before publication. These are as follows:

We are happy to learn that the Reviewer acknowledges that this manuscript has been improved. We also appreciate the constructive comments, which have been very helpful to further improve the manuscript.

comment 1: As the authors pointed out, it is quite confusing to include the "asynapsed pachytene" cells in the zygotene cell population. An alternative approach could be to include these in the pachytene cell population, or, if the authors prefer, consider them an independent population and call them abnormal/aberrant pachytene cells.

We agree that the entity of "asynapsed pachytene" was confusing verbally. In the revised version, as suggested, we considered such abnormal cells (designated as "asynapsed pachytene" in the original manuscript) as cells in an independent state and re-called them "aberrant pachytene" spermatocytes. (Lines 265-266, 270-274, and throughout the manuscript).

comment 2: The authors mention several times that "...RPA recognizes DSBs...". However, the most accurate way to say is "...RPA recognizes/binds to single stranded DNA...".

We appreciate this point. We have re-phrased relevant parts as suggested. These include "RPAs are loaded to the DSBs by binding to the single-stranded DNA generated by end resection" (Lines 196-197), "loading of RPA2 to the DSB sites"(Line 239), " RPA binding to the single-stranded DNA" (Lines 240-241), "RPA binding to single-stranded DNA" (Line 377).

comment 3: Although in image 4b and text (lines 210-215) it claims higher γ H2AX in diplotene spermatocytes grown at 37C, the quantification does not represent that. Please rectify.

Although the plots in Figure 4c may not be clear, the values of γ H2AX in diplotene grown at 34°C and 37°C were higher than those developed *in vivo*. The actual averages are 0.30, 0.47, and 0.56 for *in vivo*, 34°C and 37°C diplotene, respectively. The p-values are 0.0156 and 2.59E-5 for comparisons between *in vivo* and 34°C, and *in vivo* and 37°C, respectively.

comment 4: In lines 226-227: "...the numbers of RPA2 foci were comparable to *in vivo* samples at all the temperatures tested." – This does not apply to zygotene stage where the RPA foci number decreases significantly at 34 and 37C relative to *in vivo*. Please rectify.

We agree that this sentence may not describe the result of RPA2 foci quantification precisely. We have rewritten this part in a more precise fashion as follows (Lines 234-240).

"In explants grown at 34 °C, the number of RPA2 foci was slightly smaller than *in vivo* samples. A larger decrease was observed at 37 °C, whereas comparable numbers of RPA2 to that in *in vivo* samples were observed at 38°C (Fig. 5a-b). However, given the considerable increase in DSBs at 37 °C and the even more dramatical increase at 38°C (Fig. 4b-c), we concluded that the loading of RPA2 to the DSB sites is compromised at 37 °C and 38°C."

comment 5: Lines 228-229: "...we conclude that the DSB recognition by RPA was compromised at these temperatures." – An alternative explanation is that DSB resection is affected at higher temperatures. Please discuss this in the paper.

This is an important suggestion, which we totally agree. In revision, we have discussed that DSB resection is one of the potential primary heat-sensitive processes as follows (Line 372-378).

"The findings of this study suggest that the increase in DSBs in leptotene and zygotene is the primary cause of meiotic failure at high temperatures. This appears to be, at least in part, due to the defective assembly of the

DSB repair machinery on the DSBs, given the reduced RPA2, RAD51, and DMC1 foci. The potential underlying heat-sensitive processes include DNA resection generating single-stranded DNA, RPA binding to single-stranded DNA, and interaction between these proteins .”

comment 6: Lines 275-278 are contradicting: “...a fraction of the (fully synapsed) pachytene spermatocytes surviving at 34C and 37C harboured MLH1 foci in numbers roughly comparable with those in vivo. However, their number reduced significantly to about 20-27 at 34C, and 17-25 at 37C...”. Please clarify.

We have clarified these confusing descriptions to the following (Line 288-291):

“In contrast, a fraction of fully synapsed pachytene spermatocytes surviving at 34 °C and 37 °C harbored MLH1 foci, indicating their progression to late pachytene. However, the number of MHL1 foci reduced to approximately 20-27 at 34 °C, and 17-25 at 37 °C (Fig. 6d-g). ”.

comment 7: Lines 278-279: “Given the 19 autosome pairs and a pair of sex chromosomes, some 37C grown pachytene spermatocytes include fully synapsed chromosomes lacking MLH1 foci (Fig. 6d-e). Thus, crossover formation is compromised in late pachytene developed ex vivo, subcritically at 34C, and critically at 37C.” – perhaps, an alternative interpretation is that these pachytene spermatocytes that do not have MLH1 foci are spermatocytes that did not progress enough into pachytene given that COs (a.k.a. MLH1 foci) are known to be established in mid-to-late pachytene spermatocytes. Please discuss this in the paper. In this regard, and for future reference, many studies stain spermatocytes for the testis-specific histone H1 (H1t) to differentiate early from mid-to-late pachytene spermatocytes (e.g. Papanikos et al., 2019).

We have realized that these sentences were also confusing. The original statement seems to have confused the Reviewer as if we suggested that the spermatocytes with zero MHL1 foci might be late-pachytene spermatocytes whose MHL1 foci were totally defected by the high temperature. However, this was not what we intended to convey. Rather, we sought to say that, within each of the fully synapsed pachytene spermatocytes, while many chromosome pairs had MHL1 foci, some were devoid of the foci. To clarify this, we have rephrased it to the following (Lines 291-293):“Given the 19 autosome pairs and a pair of sex chromosomes, chromosome pair(s) lacking MHL1 foci were often observed in pachytene spermatocytes grown at 37 °C (Fig. 6e).”.

We also appreciate the suggestion of using Histone H1t to differentiate the pachytene stages, which is really valid for future studies.

comment 8: Lines 55-57: the sentence “In *C. elegans*, transient heat shock causes dysregulation of transposon expression and DNA damage in spermatocytes.” seems to be misplaced given that the next sentence (line 58) refers to testicular temperature.

To avoid confusion, we have deleted this sentence from the Introduction section. However, this information including the reference has also been provided in a paragraph evaluating the involvement of transposons in the Result section (Lines 299-301).

comment 9: The English throughout the manuscript needs to be improved as in some instances it reads a bit slang or lacks clarity (some examples are listed below).

Line 7 : ... meiotic prophase I is damaged, which offers increased DNA...

Lines 8-10: Such damaged spermatocytes cause asynapsis between homologous chromosomes, eliminated through apoptosis at the meiotic checkpoint.”

Line 215: ... spermatocytes showed higher γ H2AX levels, too.”

Line 322: Here, what causes DSBs and asynapsis does not matter...

Lines 330-331: At 34°C, prophase I progresses largely normally despite...

Line 360: ...these proteins and the cut DNA may be compromised...

The abstract also needs to be significantly improved. I suggest the authors to resort to a professional editing service.

We have re-phrased the pointed sentences, and had the manuscript edited by a professional service. The sentence in Line 322 was deleted, which we found clarifies the context effectively. Line 360 has been re-written largely in accordance with comment 5.

comment 10: Spread images are beautiful but in Figs.4b and 5 they are too small to be appreciated. Please consider expanding these into new figures.

We are happy that the Reviewer likes our nuclear spread images. As suggested, we have enlarged the photos in Fig. 5, resulting in splitting this figure as we discussed in the emails. We think that Fig. 4b may not be enlarged further given the size limit of the figures.

Reviewer #3 (Remarks to the Author):

The authors have properly addressed most of my comments, and the resubmitted version of the manuscript is significantly improved compared to the previous one. Thus, I have no objection to the publication of this study in Communications Biology.

We appreciate again the evaluation by the Reviewer. We are happy about his/her positive evaluation.

REVIEWERS' COMMENTS:

Reviewer #1 (Remarks to the Author):

The revised version of Hirano et al., is significantly improved and the authors have addressed most of my comments. It is now suitable for publication in Communications Biology.

Reviewer #1 (Remarks to the Author):

The revised version of Hirano et al., is significantly improved and the authors have addressed most of my comments. It is now suitable for publication in Communications Biology.

We appreciate again the evaluation by the Reviewer. We are happy about his/her positive evaluation.